



# Model intercomparison of COSMO 5.0 and IFS 45r1 at kilometer-scale grid spacing

Christian Zeman[1], Nils P. Wedi[2], Peter D. Dueben[2], Nikolina Ban[3], and Christoph Schär[1]

[1]Institute for Atmospheric and Climate Science, ETH Zurich, Switzerland
[2]European Centre For Medium-Range Weather Forecasts, Reading, UK
[3]Department of Atmospheric and Cryospheric Sciences, University of Innsbruck, Austria

**Correspondence:** Christian Zeman (christian.zeman@env.ethz.ch)

**Abstract.** The increase in computing power and recent model developments allow the use of global kilometer-scale weather and climate models for routine forecasts. At these scales, deep convective processes can be partially resolved explicitly by the model dynamics. Next to horizontal resolution, other aspects such as the applied numerical methods, the use of the hydrostatic approximation, and timestep size are factors that might influence a model's ability of resolving deep convective processes.

In order to improve our understanding of the role of these factors, a model intercomparison between the nonhydrostatic COSMO model and the hydrostatic Integrated Forecast System (IFS) from ECMWF has been conducted. Both models have been run with different spatial and temporal resolutions in order to simulate two summer days over Europe with strong convection. The results are analyzed with focus on vertical wind speed and precipitation.

Results show that even at around 3 km horizontal grid spacing the effect of the hydrostatic approximation seems to be

negligible. However, timestep proves to be an important factor for deep convective processes, with a reduced timestep generally allowing for higher updraft velocities and thus more energy in vertical velocity spectra, in particular for smaller wavelengths. A shorter timestep is also causing an earlier onset and peak of the diurnal cycle. Furthermore, the amount of horizontal diffusion plays a crucial role for deep convection with more diffusion generally leading to larger convective cells and higher precipitation intensities. The study also shows that for both models the parameterization of deep convection leads to lower updraft and

precipitation intensities and biases in the diurnal cycle with a precipitation peak which is too early.

## 1   Introduction

The earth's atmosphere is home to processes ranging from scales as large as the planet itself, such as the trade winds, down to scales of angstroms ($10^{-10}$ m), such as Rayleigh scattering of sunlight by an air molecule. Explicitly resolving all these processes in an atmospheric model is virtually impossible, even in the distant future. But the ever greater availability of com-

puting power allows us to at least come closer by reducing spatial resolutions in numerical weather prediction and climate models step-by-step (Schulthess et al., 2019; Neumann et al., 2019). One of the processes that can nowadays be resolved is deep convection: the rise of buoyant plumes, strong enough to break through the temperature inversions and reaching as high as to the tropopause. Given that there is enough moisture in the air, the plumes can form towering cumulonimbus clouds and cause heavy thunderstorms. On a larger scale, deep convection is an important process for the redistribution of heat, moisture,





and momentum with subsequent large impact on the general circulation in the atmosphere (Houze and Betts, 1981; Held and Soden, 2006).

Atmospheric models with a grid spacing of around 4 km and smaller have been considered to at least partially resolve deep convection (Weisman et al., 1997; Romero et al., 2001; Done et al., 2004) while models with coarser resolutions have to rely on parameterization of deep convection. One of the drawbacks of parameterized deep convection is a known phase error in

the diurnal cycle of precipitation, by being too closely coupled to the phase of solar radiation and thus peaking too early (Yang and Slingo, 2001; Betts and Jakob, 2002; Guichard et al., 2004; Dai and Trenberth, 2004), even though more recent developments by Bechtold et al. (2014) have shown some improvements in this regard. Coarse models with parameterized deep convection also tend to overestimate precipitation frequency but underestimate precipitation intensity (Dai and Trenberth, 2004; Sun et al., 2006; Stephens et al., 2010). The explicit treatment of deep convection usually leads to a better representation

of the diurnal cycle (Hohenegger et al., 2008; Dirmeyer et al., 2012; Ban et al., 2014; Pearson et al., 2014; Leutwyler et al., 2017), a better spatial representation of rainfall (Kendon et al., 2012; Prein et al., 2013), and more realistic hourly intensities of extreme precipitation events (Prein et al., 2013; Ban et al., 2014; Fosser et al., 2014; Kendon et al., 2019). More details about the benefits of kilometer-scale climate models can be found in several review articles (Prein et al., 2015; Schär et al., 2020).

Nonetheless, deep convection is not yet fully resolved with grid spacings of 1-4 km. To fully resolve deep convection,

one would require a grid spacing of around 250 m or less (Bryan et al., 2003; Lebo and Morrison, 2015; Jeevanjee, 2017). But even though the structural convergence of updrafts and clouds is not yet reached at kilometer-scale, many domain-averaged and integrated properties related to a large ensemble of convective cells (i.e. mean diurnal cycle, spatial distribution of precipitation, clouds, diabatic heating, convective transport of mass, heat and water vapour) have been shown to converge already at a grid spacing of around 4 km (Panosetti et al., 2018, 2019). This so-called bulk convergence makes the explicit treatment of deep

convection at kilometer-scale an attractive practice, as it can bring the aforementioned improvements without paying the huge computational costs associated with fully resolving deep convection. Recent work by Vergara-Temprado et al. (2020) has shown that the explicit treatment of deep convection may already be beneficial even at relatively coarse grid spacings of up to 25 km for selected metrics such as hourly precipitation statistics and the representation of the diurnal cycle over nonorographic regions.

Another subject that is often associated with weather and climate models at kilometer-scale is the use of the hydrostatic approximation in the governing equations. The hydrostatic approximation assumes the vertical accelerations to be small compared to the buoyancy force. This is normally the case when the horizontal length scale of the flow is much larger than the vertical length scale. With the hydrostatic approximation, vertical velocity can be derived from the continuity equation and thus becomes a diagnostic variable. The resulting system of equations is simpler and usually computationally less expensive to

solve, what makes it an attractive option for models as long as the hydrostatic approximation is still suitable. For example, the nonhydrostatic version of the Integrated Forecast System (IFS) model from the European Centre for Medium-Range Weather Forecasts (ECWMF) is about 80% more expensive than the corresponding hydrostatic version at a grid spacing of around 9 km (Wedi et al., 2009).



There is not really a consensus in the scientific community about the horizontal resolution at which the hydrostatic approxi-
mation is no longer suitable. For example, Ross and Orlanski (1978) performed a two-dimensional simulation of an idealized
cold front and found no big differences between the hydrostatic and the nonhydrostatic setup for a grid spacing of 20 km,
while Orlanski (1981) found significant differences for a similar case with 8 km grid spacing. According to calculations by
Daley (1988), models with a grid spacing of 25 km or smaller should already use the nonhydrostatic set of equations. But then
again, Dudhia (1993) found only little differences between the hydrostatic and the nonhydrostatic solution for a cold front
with grid spacing of 6.5 km. Kato and Saito (1995) performed idealized moist convection simulations with grid spacings of
20 km, 10 km, and 5 km and found that the hydrostatic model without parameterized deep convection overdevelops updrafts
and overestimates convective precipitation amount and area. These results were later confirmed for a real-world case from Kato
(1996). Kato (1997) recommends the use of moist convective parameterization (e.g. Manabe et al., 1965) when using a hydro-
static model with around 10 km grid spacing and the use of a nonhydrostatic model for a grid spacing of 5 km. Recent global
real-world simulations with the hydrostatic IFS at a grid spacing of 1.45 km by Dueben et al. (2020) and Wedi et al. (2021)
produced realistic results and did not show deficiencies that could be directly attributed to the invalidity of the hydrostatic
assumption at this resolution.

Several studies also primarily looked at the vertical velocities of hydrostatic and nonhydrostatic models at different resolu-
tions. A maybe counter-intuitive behavior of the hydrostatic regime is the development of too high vertical wind velocities at
resolutions where the hydrostatic assumption is no longer valid. This is due the fact that the vertical wind velocity is directly
diagnosed from the horizontal velocities and there is no nonhydrostatic process limiting the vertical mass flux. Simulations of a
squall line with horizontal grid spacings reaching from 20 km to 1 km by Weisman et al. (1997) showed the hydrostatic model
overestimating the maximum vertical velocity at grid spacings of 4 km and lower. Jeevanjee (2017) found an overestimation of
vertical velocities of the hydrostatic model at grid spacings smaller than 2 km in radiative-convective-equilibrium simulations
over sea with grid spacings ranging from 16 km to 0.0625 km. Dueben et al. (2020) performed global simulations with IFS
using the hydrostatic and nonhydrostatic equations at 1.45 km grid spacing where the updraft velocities were quite similar
when using a timestep of 30 s.

Not only the system of equations, but also the applied numerical methods are important when it comes to understanding
the model behavior. The two models used for this study are very different in this regard: While the hydrostatic IFS model
is a spectral model with a semi-Lagrangian semi-implicit scheme, the nonhydrostatic COSMO model is a Eulerian model
with a split explicit scheme in the horizontal and a implicit scheme in the vertical dimension. These differences in design
have direct implications on the conditions for numerical stability and the associated timestep of the models. Thanks to the
semi-Lagrangian treatment of advection, the timestep in IFS is not limited by the Courant–Friedrichs–Lewy (CFL) condition
(Courant et al., 1928) but by the Lipschitz condition. The Lipschitz condition requires the timestep to be smaller than the
reciprocal of the absolute maximum value of the wind shear at each direction (Pudykiewicz et al., 1985; Staniforth and Côté,
1991). It ensures that the trajectories do not intersect each other (Smolarkiewicz and Pudykiewicz, 1992) and is less restrictive
regarding timestep than the CFL condition, allowing an atmospheric model to remain stable and deliver accurate results even
with CFL numbers higher than 4 (Staniforth and Côté, 1991). This implies that IFS can be run with a rather long timestep





and still remain stable, even though the CFL condition might be violated at some locations with high wind speed. In contrast,
COSMO uses an Eulerian explicit approach for horizontal advection and thus the timestep has to be small enough to not violate
the horizontal CFL condition at any location in order to guarantee stability.

Compared to the many studies addressing spatial resolution in atmospheric models, the sensitivity to temporal resolution has
received relatively little attention. Several studies identified timestep as a very important factor when it comes to precipitation
patterns (Williamson and Olson, 2003), precipitation intensity (Mishra and Sahany, 2011), or tropospheric circulation (Jung
et al., 2012). However, these studies were carried out with relatively coarse resolution ($\Delta x > 100\,\mathrm{km}$) with parameterized
deep convection and are therefore hardly comparable to the resolutions used in this study. Fuhrer et al. (2018) recommends a
timestep smaller than 40 - 60 s at around 1 to 2 km grid spacing. The global simulations with IFS from Dueben et al. (2020)
with a 1.45 km grid spacing showed improvements in the representation of deep convective processes for both, the hydrostatic
and nonhydrostatic version when reducing the timestep from 120 s to 30 s. It is probably difficult to give a generally valid
recommendation for timestep size, as many different processes are affected by it. Next to the dynamics, the parameterization
of subgrid-scale processes, its call frequency, and the type of coupling to the model dynamics (see for example  Ubbiali et al.,
2021) also have to be considered. For instance, Barrett et al. (2019) performed idealized simulations using COSMO with 1 km
grid spacing and found a 53% reduction in precipitation with a two-moment microphysics scheme when the timestep was
increased from 1 s to 15 s. These changes were attributed to the timestep dependence of the amount of supersaturation with
respect to liquid in strong updrafts and the corresponding sensitivity of the cloud microphysics parameterization to this value
in combination with the sequential-update splitting coupling. In the current study we use bulk microphysics schemes, which
have a much smaller sensitivity with respect to the timestep.

Deep convection is a dynamical process that is often happening very locally, involving only a few grid points in kilometer-
scale models. The dynamics and concentration of moist variables at such scales are largely affected by diffusion. Diffusion
may serve many purposes, such as eliminating numerical noise, increasing model stability, absorbing vertically propagating
gravity waves at the model top, or also emulating cumulative effects of unresolved subgrid-scale processes (see Jablonowski
and Williamson, 2011, for an overview of diffusion). Next to implicit diffusion, which is inherently caused by the numerical
methods, most models apply some form of explicit diffusion. A significant amount of diffusion is also caused by subgrid-
scale parameterizations and orography filtering (Malardel and Wedi, 2016). All these aspects can lead to very different model
behavior in terms of dissipation, which then might again influence deep convection.

Ricard et al. (2013) conducted a case study over southwest France with the nonhydrostatic limited-area model AROME
(Météo-France), which uses the same dynamics as the nonhydrostatic version of IFS, at 2.5 km grid spacing in order to deter-
mine the influence of horizontal diffusion on convective cells. They have compared the results to the Eulerian research model
Meso-NH (Lafore et al., 1998) and found that AROME develops larger convective cells than Meso-NH with a tendency of the
cells to structure into circular patterns with too strong outflow at the surface induced by cooling from precipitation evapora-
tion, especially with additional explicit diffusion. In order to prevent too high precipitation intensities and unrealistic divergent
winds at the edges of the cold outflow, Malardel and Ricard (2015) introduced a correction of the interpolation weights of the
semi-Lagrangian scheme for IFS, AROME, and HARMONIE (HIRLAM consortium), which all use the same dynamics. These





new interpolation weights improve the conservation property of the scheme, effectively increase diffusion in the convergent parts of the flow and reduce diffusion in the divergent parts of the flow. They performed idealized and real-case experiments of convective systems with the new interpolation weights, which lead to a significant reduction of extreme precipitation with the new interpolation weights. While the experiments shown in the paper were performed with the nonhydrostatic version of the dynamics of IFS, the operational hydrostatic version showed the same improvements.

In order to improve our understanding about the role of some of the aforementioned factors in the representation of deep convection, we here present a model intercomparison between COSMO and IFS, addressing the following key questions:

1. What are the main differences between COSMO and IFS in the representation of deep convective precipitation? How do the precipitation patterns, precipitation intensities, and the diurnal cycle of precipitation look like at different resolutions, and how do they compare with observations?

2. Can we detect significant differences due to the use of the hydrostatic versus nonhydrostatic dynamics in IFS and COSMO, respectively?

3. What is the effect of the timestep on deep convection in the different models? Are there any disadvantages of using a large timestep for simulations with explicit treatment of deep convection?

4. How does horizontal diffusion affect deep convection? Do the two models show differences in the representation of deep convection that can be accounted to differences in horizontal diffusion?

It has to be mentioned, that comparing two so fundamentally different and complex models in simulating a real-world case makes it intriguing but also very difficult to confidently attribute any disparities in the examined results to a specific process or its associated handling in the model. Nevertheless, some assumptions can be made based on findings in previous studies and our knowledge of the different model properties. We would also like to emphasize that this study is not intended to be a performance comparison of the two models. The models have not been specifically tuned for the respective resolution setups and, with only two days of hourly data, the sample size is too small to draw any conclusions regarding the general forecast quality. Differences in the results are expected, as the specific weather situation under consideration was difficult to predict and the general setup of the models is different. In particular, IFS is initialized from its own analysis and then run globally, while COSMO is driven by the IFS operational analysis. So the aim of the study is mainly to show differences and similarities of these two very distinct modelling approaches with different configurations in their treatment of deep convective processes, and through this extend our knowledge on this subject.





## 2 Methods and data

### 2.1 Model description

#### 2.1.1 COSMO

The Consortium for Small-scale Modelling (COSMO) model (Baldauf et al., 2011) has been originally developed for numerical
weather prediction, but has been extended to also run in climate mode (Rockel et al., 2008). COSMO is a regional model and
operates on a grid with rotated latitude–longitude coordinates. It uses a split explicit third-order Runge-Kutta discretization
(Wicker and Skamarock, 2002) in combination with a fifth-order upwind scheme for horizontal advection, and an implicit
Crank-Nicholson scheme for vertical advection. Parameterizations used in this version include subgrid-scale orography (SSO)
by Lott and Miller (1997), a radiation scheme based on the $\delta$-two-stream approach (Ritter and Geleyn, 1992), a single-moment
cloud microphysics scheme (Reinhardt and Seifert, 2006), a turbulent kinetic energy based parameterization for the planetary
boundary layer (Raschendorfer, 2001), an adapted version of the convection scheme by Tiedtke (1989), and a multi-layer
soil model with a representation of groundwater (Schlemmer et al., 2018). Explicit horizontal diffusion is applied by using
a monotonic 4th-order linear scheme acting on model levels for wind, temperature, pressure, specific humidity, and cloud
water content (Doms and Baldauf, 2018). An orographic limiter helps avoiding excessive vertical mixing around mountains.
For the standard experiments in this paper, the explicit diffusion from the monotonic 4th-order linear scheme is set to zero.
However, we apply Smagorinsky diffusion (Smagorinsky, 1963) to the horizontal wind components for all experiments in
order to enhance the numerical stability of the scheme in the presence of horizontal shear instabilities. For this project we
use a refactored version of COSMO 5.0, which is able to run on hybrid GPU-CPU architectures (Fuhrer et al., 2014). The
model extension was a joint effort between MeteoSwiss, the ETH-based Center for Climate Systems Modeling (C2SM), and
the Swiss National Supercomputing Center (CSCS). It has been used for climate studies with 2.2 km grid spacing over Europe
(Ban et al., 2014; Leutwyler et al., 2017) and is also capable of running on a near-global domain at this resolution (Fuhrer
et al., 2018).

#### 2.1.2 IFS

The Integrated Forecasting System (IFS) is the model used by the European Centre for Medium-Range Weather Forecasts
(ECWMF) for its daily data assimilation and subsequent global forecasts. It is a hydrostatic model but can also be run using a
nonhydrostatic extension which originally has been developed for the ARPEGE/Aladin models (Bubnová et al., 1995; Bénard
et al., 2010). IFS is a spectral transform model where temperature, wind, and surface pressure are represented in spectral
space with spherical harmonics basis functions, transformed at every timestep to a corresponding grid point space on a cubic-
octahedral reduced Gaussian grid (Wedi, 2014; Malardel et al., 2016). Notably, all water substance variables only exist in
gridpoint space. Semi-Lagrangian advection, physical parameterizations, and nonlinear terms are calculated in grid point space.
The horizontal gradients and the Laplacian operator for horizontal wave propagation are then efficiently calculated in spectral
space. The transformation between grid point space and spectral space is done by a Fast Fourier Transformation (FFT) in



longitude and a (Fast) Legendre Transformation (FLT) in latitude. The spectral transforms do not scale linearly with the number of grid points and also require global communications, which means that at very high resolution the spectral transforms

become a computational bottleneck of the model (Wedi et al., 2013; Schär et al., 2020). One of the reasons why the global IFS is still mostly run in hydrostatic mode is that the nonhydrostatic version uses a predictor-corrector approach that leads to more spectral transforms per timestep and is therefore substantially slower. Recently, a nonhydrostatic core for IFS based on a finite-volume discretization (IFS-FVM) has been developed (Kuhnlein et al., 2019). IFS-FVM does not require spectral transforms and achieves a higher strong scaling computational efficiency compared to the spectral model at higher resolutions.

However, this paper focuses on the hydrostatic spectral model version of IFS which is still very competitive in terms of time-to-solution even at kilometer-scale (e.g. Kuhnlein et al., 2019; Schulthess et al., 2019; Dueben et al., 2020; Wedi et al., 2021). IFS uses an adapted version of the convection scheme by Tiedtke (1989) with improvements regarding tropical variability (Bechtold et al., 2008) and diurnal cycle (Bechtold et al., 2014). Other parameterizations include a Monte-Carlo Independent Column Approximation (McICA) for radiation (Barker et al., 2008; Hogan et al., 2017) and the land surface hydrology scheme

HTESSEL (Balsamo et al., 2011). IFS applies explicit diffusion to the prognostic variables in spectral space (temperature, wind, surface pressure) with an operator that mimics spectral viscosity after Gelb and Gleeson (2001). Furthermore, some diffusion comes implicitly from the interpolation required by the semi-Lagrangian scheme in grid point space (notably for the water variables and tracers), as well as the spectral truncation due to the transformation from grid point space to spectral space, acting like a $4\Delta x$ spectral filter in case of a cubic grid. The simulations for this project were performed with the global

atmospheric model of the IFS based on Science Version 45r1. IFS documentation of the different model cycles can be found on the ECMWF website (https://www.ecmwf.int/en/publications/ifs-documentation, last access: 1 February 2021).

## 2.2 Numerical experiment and model setup

### 2.2.1 Model intercomparison

The simulations cover two days from 29 May 2018 00:00 UTC to 30 May 2018 with heavy thunderstorms over Europe. Both

models use one day lead time (28 May) and are initialized with ECMWF operational analysis data at a horizontal grid spacing of $\sim 9$ km. Since COSMO uses a different soil model (Schlemmer et al., 2018), the soil in COSMO was initialized with an average value from May/June after a 5-year spinup with COSMO 12 km by Vergara-Temprado et al. (2021). IFS is run globally, whereas COSMO is run regionally with the ECMWF operational analysis data as lateral boundary conditions on a limited area domain ranging from $361 \times 361$ ($\Delta x = 12$ km) to $1542 \times 1542$ ($\Delta x = 2.2$ km) grid points. The domain for this study is shown

in Fig. 1. Simulations have been performed for a range of horizontal grid spacings for both models. The grid spacings used with COSMO are $\Delta x = 12$ km, 4.4 km, and 2.2 km, whereas the grid spacings used with IFS approximately correspond to $\Delta x = 9$ km, 4.5 km, and 2.9 km. Both, COSMO and IFS usually use deep convection parameterization for coarser resolutions such as $\Delta x = 12$ km for COSMO or $\Delta x = 9$ km for IFS. However, for this study the deep convection parameterization is switched off per default and only one simulation for each model has been performed at the coarsest resolution with deep

convection parameterization on. In order to test the impact of timestep in the respective simulations, each horizontal resolution





**Table 1.** Model description and setup.

|  | COSMO | IFS |
| --- | --- | --- |
| **Numerics:** | Split Explicit | Semi-Lagrangian and Semi-Implicit |
| **Vertical Velocity:** | Nonhydrostatic | Hydrostatic |
| **Horizontal Discretization:** | Rotated Lat/Lon | Spectral and Reduced Gaussian, octahedral |
| **Resolution Setups:** | $\Delta x = 12\,\mathrm{km}\ (0.11°),\ \Delta t \in \{90\,\mathrm{s}, 40\,\mathrm{s}\}$ | $\Delta x = 9\,\mathrm{km}\ (\mathrm{TCo}\ 1279),\ \Delta t \in \{450\,\mathrm{s}, 240\,\mathrm{s}\}$ |
|  | $\Delta x = 4.4\,\mathrm{km}\ (0.04°),\ \Delta t \in \{40\,\mathrm{s}, 20\,\mathrm{s}\}$ | $\Delta x = 4.5\,\mathrm{km}\ (\mathrm{TCo}\ 2559),\ \Delta t \in \{240\,\mathrm{s}, 120\,\mathrm{s}\}$ |
|  | $\Delta x = 2.2\,\mathrm{km}\ (0.02°),\ \Delta t \in \{20\,\mathrm{s}, 10\,\mathrm{s}\}$ | $\Delta x = 2.9\,\mathrm{km}\ (\mathrm{TCo}\ 3999),\ \Delta t \in \{120\,\mathrm{s}, 60\,\mathrm{s}\}$ |
| **Vertical Levels:** | 60 (up to $\sim 23.5\,\mathrm{km}$)[c] | 137 (up to $\sim 80.5\,\mathrm{km}$)[c] |
| **Convection Param.:** | Deep Convection off[a] | Deep Convection off[b] |
|  | Shallow Convection on | Shallow Convection on |

[a]For $\Delta x = 12\,\mathrm{km}$, $\Delta t = 90\,\mathrm{s}$ also one run with deep convection parameterization on

[b]For $\Delta x = 9\,\mathrm{km}$, $\Delta t = 450\,\mathrm{s}$ also one run with deep convection parameterization on

[c]The vertical spacing is the same for all resolution setups.

setup with explicit deep convection has also been run with a smaller than usual timestep. Some model properties and the respective configurations are listed in Table 1.

### 2.2.2 Horizontal diffusion experiment

For this experiment, COSMO has been run for the same case as above, but with a varying amount of explicit horizontal
diffusion. This will give us some idea about the influence of horizontal diffusion on the model results and might explain some characteristic differences between IFS and COSMO. In COSMO, the default diffusion coefficient for the 4th-order diffusion is $\alpha_4 = (\Delta x/2)^4/\Delta t$ and acts on wind, temperature, pressure, specific humidity, and cloud water content. This coefficient can be multiplied with a factor, which we will hereafter call `diff`, in order to apply more or less smoothing to the mentioned variables. A value of `diff = 1` means that the diffusion coefficient remains unchanged and corresponds to the default value
$\alpha_4$. Any value of `diff` smaller than one decreases the explicit diffusion coefficient and any value larger than one increases explicit 4th-order diffusion strength. In our default setup for the intercomparison, COSMO has been run with no explicit 4th-order linear horizontal diffusion, which means `diff` was set to zero. For this experiment, the 2.2 km setup with a timestep of $\Delta t = 20\,\mathrm{s}$ has been used, but with numbers for `diff` ranging from 0 to 4 with an increment of 0.5.

### 2.3 Observations

Three datasets are used for the evaluation of the model results: IMERG, RADKLIM, and IDAWEB. Comparing model results with observational data is a difficult undertaking. Next to the differences in spatial sampling (i.e. point measurement vs. grid cell averages), observations also suffer from several deficiencies (see below) and therefore different observational datasets





often provide substantially different results, which is also the case in this study. Thus, observations should only be taken as a point of reference and not the absolute truth.

### 2.3.1 IMERG

The Integrated Multi-satellitE Retrievals for GPM (IMERG) dataset (Huffman et al., 2019b) provides worldwide, half-hourly precipitation data on a $0.1° \times 0.1°$ grid by using a set of algorithms to combine satellite data and rain gauge observations into one product (Huffman et al., 2019a). IMERG incorporates satellite data from as many satellites as possible, i.e. not only the ones under the direction of the Global Precipitation Measurement (GPM) mission, in a flexible framework. The satellite data consists of passive microwave (PMW) sensors from various low-Earth-orbit platforms and infrared (IR) estimates from geosynchronous-Earth-orbit satellites, as well as active radar data from the GPM satellites. The rain gauge data stems from the Global Precipitation Climate Centre (GPCC) which is operated by the German Weather Service (DWD, Deutscher Wetter Dienst). The specific product that has been used by IMERG for the time period of this case study is the GPCC Monitoring Product V6 (Schneider et al., 2018). This product is based on monthly SYNOP and CLIMAT data from 7000 - 9000 rain gauges worldwide. IMERG adjusts the accumulated monthly precipitation totals from GPCC with a gauge correction algorithm by Legates and Willmott (1990) and then calibrates the gridded multi-satellite estimate with these values. For this study, the Final version of IMERG has been used and the half-hourly measurements were added up to hourly values in order to be consistent with the model output frequency.

### 2.3.2 RADKLIM

RADKLIM (Radarklimatologie) is a radar-derived and gauge-adjusted precipitation product from the German Weather Service (DWD, Deutscher WetterDienst) that works on a $1100 \times 900$ grid over Germany with 1 km grid spacing (Winterrath et al., 2017). It uses measurements from 17 C-band weather radars (for the evaluated period only 16 radars have been in use) and approximately 2000 rain gauge stations. The method is based on the disaggregation of daily precipitation estimates from raingauges into hourly values using radar-based estimates (Paulat et al., 2008; Wüest et al., 2010). The specific product that has been used for this study is the RW product, which uses the weighted mean of two different gauge calibration methods, from the reanalysis version 2017.002. RADKLIM delivers hourly accumulated precipitation values for the hour from (hh-1):50 to hh:50. This represents a slight shift compared to the model and averaged IMERG outputs, which are available for the (hh-1):00 to hh:00 intervals. This 10-minute time shift is neglected in this study. RADKLIM works very similar to RADOLAN (Bartels et al., 2004), but unlike RADOLAN it is not a real-time product. RADKLIM includes more rain gauge stations for the calibration ($\sim 2000$ compared to $\sim 1300$) and also possesses a more sophisticated radar artefact correction process. Therefore, RADKLIM should deliver more accurate values than RADOLAN.

Radar-based estimates of rainfall allow for a high resolution in space and time, but they are also associated with some uncertainties. Sources of errors include cluttering from other objects, attenuation, variability of the relation between reflectivity and rainfall rate (Z-R relation), beam blockage, range degradation, vertical variability of the precipitation system (i.e. bright band), and vertical air motions that increase or decrease raindrop fall speed (Villarini and Krajewski, 2010). RADKLIM uses a





sophisticated radar artefact correction process which, together with the rain gauge calibration, should minimize the uncertainty due to such artefacts. Nevertheless, some artefacts might still exist which has to be kept in mind when using the data.

An intercomparison between RADOLAN, which is very similar to RADKLIM, and IMERG can be found in Ramsauer et al. (2018), where they have found total precipitation in IMERG to be generally higher than in RADOLAN but lower in moun-
tainous regions. This underestimation of precipitation by IMERG at higher altitudes, opposing to the general overestimation in flatter terrain, was also found in studies by Wang et al. (2019) who evaluated IMERG with a dense rain gauge station network in Lhasa. They also showed that the performance of IMERG overall decreases with increasing elevation.

### 2.3.3   IDAWEB

IDAWEB is a web-portal operated by MeteoSwiss which provides hourly precipitation measurements from roughly 1000 rain
gauges from different institutions in Switzerland. Unlike IMERG or RADKLIM, it is not a gridded dataset. But due to its high density and relatively homogenous distribution of stations it provides a good estimate of rainfall in Switzerland. The rain gauges are distributed throughout various altitude levels with the lowest one in Switzerland being on 200 m (Isole di Brissago) and the highest one on 3294 m (Piz Corvatsch). With close to 200 rain gauges at an elevation of over 1500 m, IDAWEB also provides decent coverage at high altitude in the Alpine region. IDAWEB also incorporates rain gauge measurements from a
few stations outside of Switzerland, but as most of them are quite isolated, these stations were ignored for this study.

Like satellite-based or radar-based products, also rain gauge observations involve uncertainties. They suffer from various errors such as evaporation, splashing, and most importantly wind effects which usually result in a low bias. The mean undercatch for Switzerland in summer is estimated to be 7% with exposed stations having roughly twice the bias as well protected sites (Sevruk, 1985; Richter, 1995). A good overview of errors and error correction can be found in Sevruk (2005). While IMERG
uses a gauge correction algorithm by Legates and Willmott (1990), no comparable gauge correction algorithm is applied for RADKLIM or the IDAWEB station data. However, the undercatch for heavy summer precipitation is assumed to be rather small and incorporating such possible observation errors into this analysis would be beyond the scope of this study.

## 3   Results

### 3.1   Model intercomparison

### 3.1.1   Precipitation pattern

We start by showing an example of the spatial precipitation distribution. Figure 1 shows accumulated hourly precipitation between 17:00 and 18:00 UTC on 29 May 2018 from different model runs and the multi-satellite product IMERG. While the location of precipitation is generally similar to the observations, there are distinct differences visible between COSMO and IFS. Most obvious is the larger amount of light precipitation in IFS compared to COSMO. Additionally, the cell structure
in COSMO is more fine-grained than in IFS. These two model characteristics hold true throughout all resolutions and can also be seen in Fig. 2, which shows the precipitation patterns at the same time for the RADKLIM domain. Both runs with





parameterized deep convection clearly produce less heavy precipitation than the ones with explicit deep convection. Especially for the RADKLIM domain in Fig. 2, COSMO (PAR) produces much less precipitation, which is partially due to a shift in timing (precipitation falls too early in the course of the day), but also due to a generally strong underestimation of precipitation over Germany with this setup, especially for the first day (see Fig. 1 and also Table 2).

When looking at the observations only, RADKLIM and IMERG agree well on the location of the precipitation. There are, however, visible differences in intensity and spatial extent. While some of these differences might come from the different measurement and processing techniques, differences will also be caused by the much higher spatial resolution of RADKLIM.

### 3.1.2 Precipitation intensity

The cumulative frequencies of hourly precipitation within the European domain for all 48 hours are depicted in Fig. 3. Both models show an increase in frequency of precipitation ($> 0.1\,\mathrm{mm\,h^{-1}}$) but a decrease in heavy precipitation events if deep convection is parameterized. IFS produces more light precipitation than COSMO in all configurations. While the frequency of medium and heavy precipitation remain relatively constant for the COSMO runs with explicit deep convection, IFS shows increasing medium and heavy precipitation with decreasing grid spacing and timestep. It can be assumed that this increase is not so much caused by the refined horizontal resolution but rather by the smaller timestep. This becomes obvious when one compares the IFS configurations with the same timestep: The runs with the same timestep show striking similar cumulative frequencies for different horizontal grid spacings (9 vs. 4.5 km and 4.5 vs. 2.9 km). There are three possible explanations for this strong dependence of IFS on timestep:

(a) It could be that the original timestep in IFS are too large to properly represent deep convective processes associated with such high vertical wind velocities (see also chapter 3.1.5). For instance, with a 120 s timestep (IFS 2.9 km) and assuming a mid-tropospheric vertical wind speed of $20\,\mathrm{m\,s^{-1}}$, an airparcel would traverse the troposphere in merely 3-4 time steps. The large timestep thus implies inaccuracies, as the airparcel's trajectory and its forcing by diabatic heating cannot be fully accounted for. So while the semi-Lagrangian scheme prevents the model from developing instabilities, the large timestep will likely affect the convective mesoscale dynamics, truncate extreme updrafts and thus allow less heavy precipitation events. As COSMO does not show such a timestep sensitivity, one could argue that the timestep in the COSMO simulations is already small enough and thus has no significant effect on convective mesoscale dynamics. However, also COSMO develops higher vertical velocities with a smaller timestep (see Sect. 3.1.5), even though a bit less pronounced than IFS. So it is hard to imagine that this truncation of vertical updrafts is the only reason for the strong timestep dependence of precipitation in IFS.

(b) By halving the timestep, any subgrid scale parameterization scheme will be called twice as often, which may affect precipitation. Notably, paramerizations such as cloud microphysics, shallow convection, or vertical mixing could experience timestep sensitivity which could affect convective processes. Barrett et al. (2019) shows a strong timestep sensitivity of total precipitation in an idealized setup with COSMO in combination with a two-moment microphysics scheme (see above). For our simulations, COSMO and IFS both use a single-moment scheme which show little timestep sensitivity of total precipitation but some sensitivity regarding precipitation location and magnitude in Barrett et al. (2019). While we do not see an impact in



**Figure 1.** Accumulated hourly precipitation between 17:00 and 18:00 UTC on 29 May 2018 over the European domain. The top left panel shows precipitation obtained from the multi-satellite product IMERG which are provided on a $0.1° × 0.1°$ grid. The other panels show results from model runs, where the first number corresponds to the approximate grid spacing and the number in parenthesis is the respective timestep. The term PAR denotes that the respective runs used parameterized deep convection.

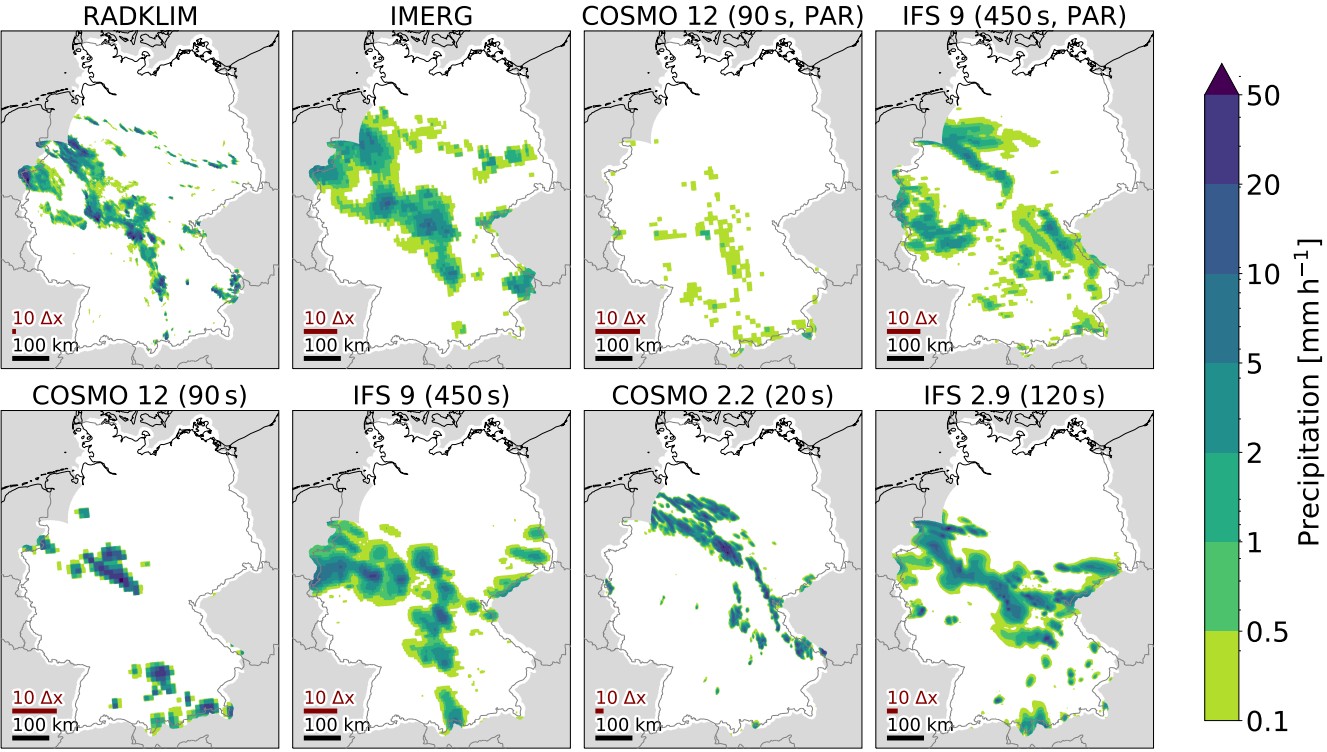

**Figure 2.** Accumulated hourly precipitation interpolated to the RADKLIM domain over Germany. While IMERG and the results from the model runs show precipitation from 17:00 to 18:00 UTC on 29 May 2018, the corresponding RADKLIM interval is 16:50 to 17:50 UTC (see Sect. 2.3).

COSMO, timestep sensitivity of subgrid scale parameterization could affect IFS, where the absolute differences in timestep size are generally larger.

(c) One possibility that has been investigated was the sensitivity of the interpolation error in the semi-Lagrangian scheme to timestep. In semi-Lagrangian schemes, the accumulation of errors is also a function of the timestep and the error of the spatial interpolation procedure (Bonaventura, 2004). Halving the timestep leads to twice as many interpolations, which can potentially

increase the total interpolation error, lead to more damping and thus increase the amount of implicit diffusion. In Sect. 3.2, it is shown that more diffusion leads to more heavy precipitation events, which could thus explain the timestep sensitivity in IFS. However, we do not see increased damping in IFS with a smaller timestep. This becomes obvious when looking at the spectra of horizontal kinetic energy in Fig. 8, where IFS does not show stronger dissipation for the very small wavelengths with a reduced timestep. The absence of such an increase in damping with smaller timestep can be explained with the influence

of timestep on departure point in semi-Lagrangian schemes. A smaller timestep will yield different departure points and thus also different damping from the quasi-cubic interpolation in IFS (see Sect. 13.6.4 in Lauritzen et al., 2011, for a more detailed discussion). We therefore consider this possibility to be less relevant when compared to (a) and (b).





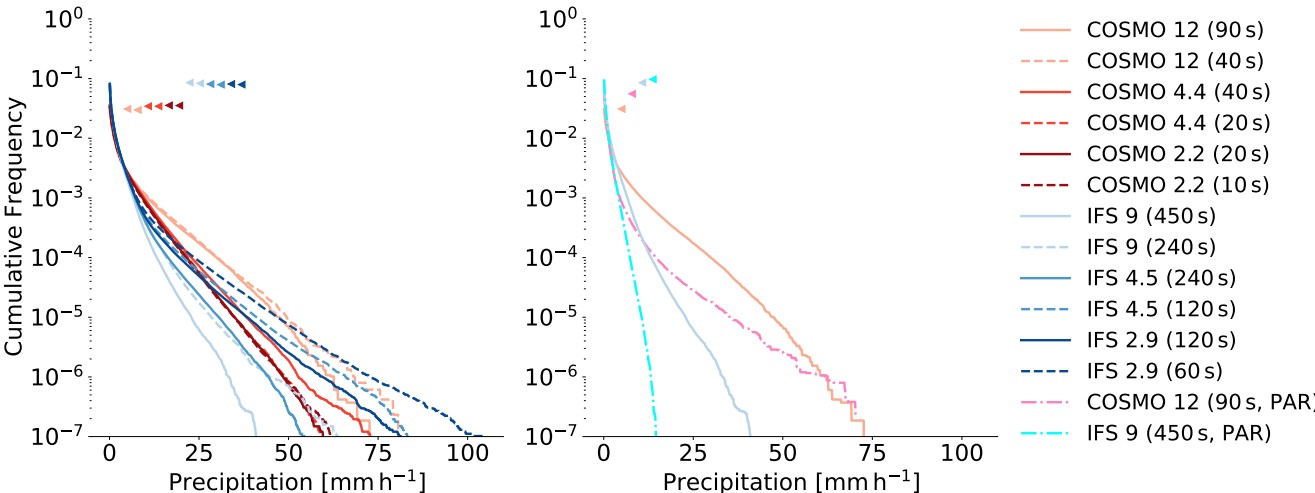

**Figure 3.** Cumulative frequency of accumulated hourly precipitation over the European domain for the whole 48 hours period. The left panel shows all run with explicit deep convection and two different timesteps per spatial resolution. The right panel shows the coarser runs with parameterized (PAR) and with explicit deep convection. The triangles indicate the frequency of precipitation $\geq 0.1\,\mathrm{mm\,h^{-1}}$ of the respective simulations.

Presumably, the timestep dependency in IFS stems from a combination of (a), (b), and (c). But getting a better insight into the respective role of these factors would require further studies and is beyond the scope of this work.

Fig. 4 shows the cumulative frequencies of precipitation for the RADKLIM domain and the IDAWEB stations. For the RADKLIM domain, the model output and IMERG data has been interpolated to the RADKLIM grid and for the IDAWEB stations, a nearest neighbor interpolation has been performed. Both RADKLIM and IDAWEB show a lower frequency of precipitation ($\geq 0.1\,\mathrm{mm\,h^{-1}}$) but a higher frequency of heavy precipitation than IMERG. The higher frequency of precipitation in IMERG compared to the other observational datasets is consistent with systematic biases of the respective analysis (Ramsauer

et al., 2018; Wang et al., 2019). In the RADKLIM domain, the model runs show a similar behavior as for the whole European domain. For the IDAWEB stations it is a bit more difficult to see a clear pattern, most likely due to the small sample size in combination with the nearest neighbor interpolation. For example, the two COSMO 2.2 km simulations show big differences in heavy precipitation. This can most probably be attributed to differences in precipitation location and subsequent different intensities obtained by the nearest neighbor interpolation, as we do not see such a behavior for the whole European domain or

the gridded RADKLIM dataset. However, both datasets, IDAWEB and RADKLIM, show that the runs with parameterized deep convection fail to produce the medium-to-heavy precipitation associated with deep convection. The cumulative frequencies of the model runs with explicit deep convection usually lie within the range of the observational values and therefore none of these configurations seems to produce unreasonable values. In general, the distribution of the runs with higher resolutions are closer to the observations.

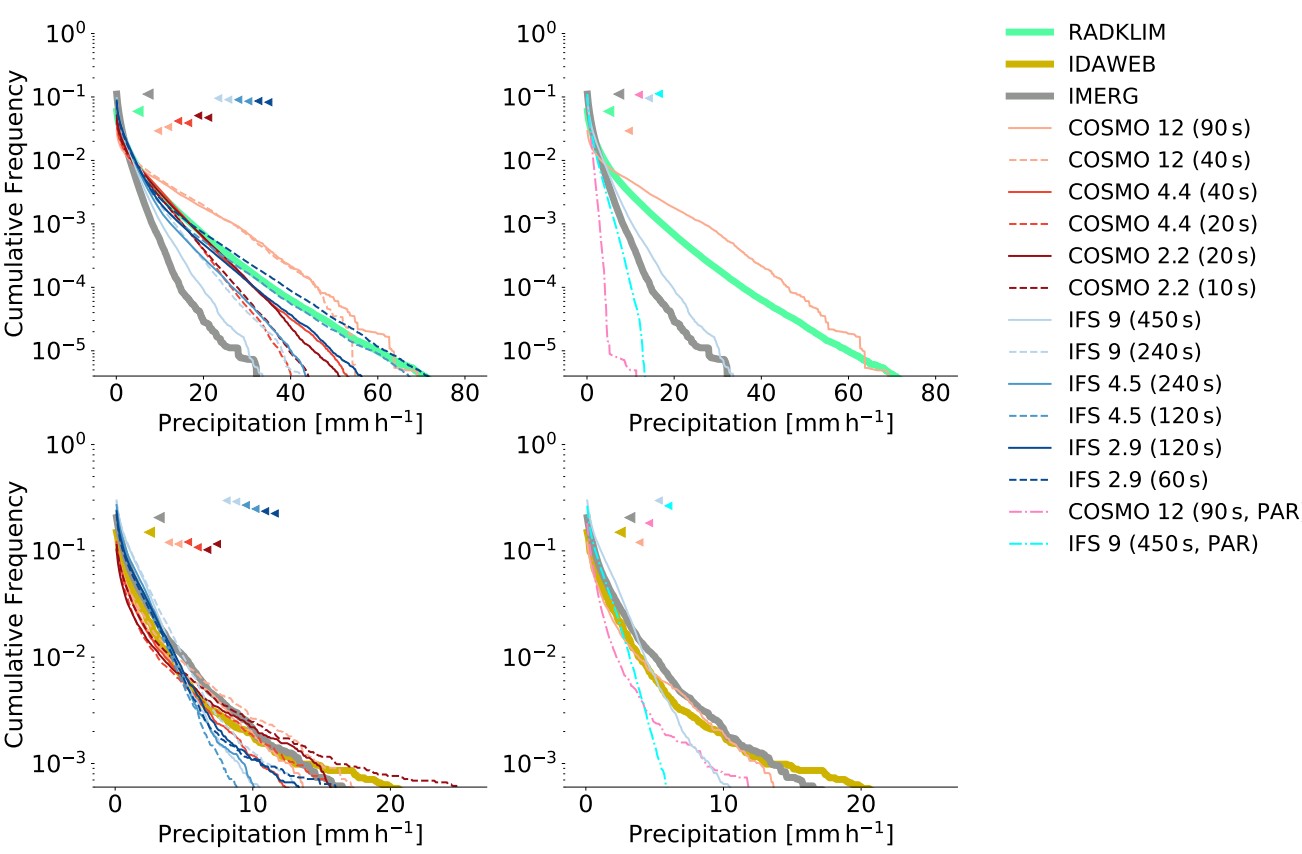

**Figure 4.** As Fig. 3 but for the RADKLIM domain (top) and IDAWEB station data (bottom). Note the different scales for the two domains.

### 3.1.3 Diurnal cycle over land

Several studies have already shown that parameterized deep convection leads to a premature diurnal cycle in COSMO (Ho-
henegger et al., 2008; Ban et al., 2014; Leutwyler et al., 2017; Vergara-Temprado et al., 2020). Figure 5 shows that this also
applies in this study. Compared to COSMO, the phase of IFS with parameterized deep convection is shifted a bit towards
the later hours, which could be a result of the improvements from Bechtold et al. (2014) to the parameterization scheme by
Tiedtke (1989). But still, both IFS and COSMO show a significant phase shift with paramterized deep convection. Convective
precipitation lasts longer in IFS than in COSMO for all configurations. Compared to the observations from IMERG, convective
precipitation in IFS lasts too long for both days, whereas in COSMO it seems to be about right for the first day but too short for
the second day. A very interesting aspect is the dependence of the diurnal cycle on spatial and temporal resolution. All runs,
except the ones that already use a rather small timestep (COSMO 4.4 and 2.2), show an earlier development and also decay of
convective precipitation when reducing the timestep. There are some signs of convergence for COSMO at 4.4 km grid spacing
and also for IFS, where the 4.5 km run with $\Delta t = 120$ s and the two 2.9 km runs are quite close. This convergence of the

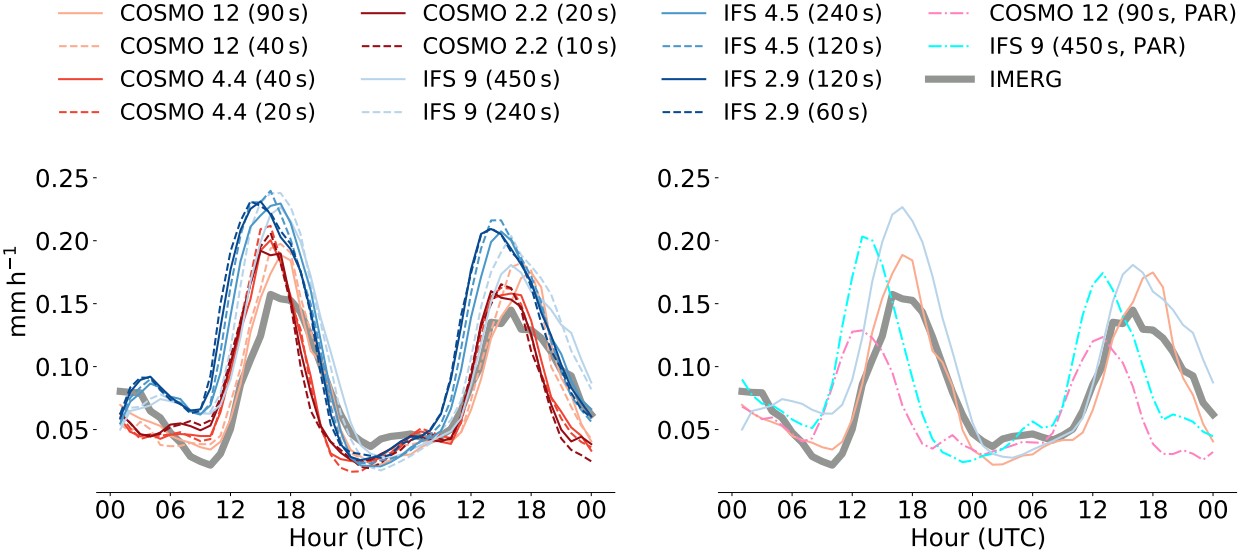

**Figure 5.** Diurnal cycle of precipitation over land in the European domain from 29 May 2018 01:00 UTC to 31 May 2018 00:00 UTC. The plot shows accumulated hourly precipitation, meaning that at 29 May 2018 01:00 UTC it shows the precipitation accumulated from 29 May 2018 00:00 UTC to 29 May 2018 01:00 UTC. The left panel shows all run with explicit deep convection and two different timesteps per spatial resolution. The results from the runs with the shorter timestep are represented by the dashed lines. The right panel shows the coarser runs with parameterized (PAR) and with explicit deep convection. Both panels also show the values from the Integrated Multi-Satellite Retrievals for GPM (IMERG) dataset for our domain.

diurnal cycle around 4 km is somewhat consistent with findings from from Langhans et al. (2012) and Panosetti et al. (2019), who simulated 9 convective summer days with COSMO over the Alps and Germany with grid spacings ranging from 8.8 km to 550 m. In their studies, the peak of the diurnal cycle then shifts back, again more towards the evening, with 1.1 km and 550 m.

It is possible that, by increasing the resolution even further, a similar effect could be seen in this case. The shift of the diurnal cycle with higher temporal resolution for the explicit COSMO 12 km is also consistent with the results from Panosetti et al. (2019), who found a shift of the 8.8 km run but none for the 4.4 km run by using $\Delta t = 5$ s instead of the original timestep of 80 s and 40 s respectively. It is not entirely clear what causes this dependence of the diurnal cycle on temporal resolution. It could be that a large timestep leads to some truncation in the dynamics and thus to a delay in inhibition. But it is also likely

that this is related to the changed frequency of application of the subgrid-scale paremeterizations. A detailed analysis of the causes for this behavior would certainly be an interesting subject for further studies.

For COSMO, Baldauf et al. (2011) has shown a high sensitivity of convection initiation to the Blackadar length scale. But as the Blackadar length scale has been kept constant for all COSMO configurations ($l_\infty = 150$ m) and as the same effect regarding resolution can be seen in IFS, which uses a different parameterization for turbulent transport (ECMWF, 2018), it can be ruled

out as a factor in this case.





**Table 2.** Mean precipitation per 48 hours in the different domains. The bold numbers represent the reference values of the respective domain that have been used to calculate the differences in percent.

| | European Domain[a] | European Domain (Land)[a] | RADKLIM Domain[b] | IDAWEB Stations[c] |
|---|---|---|---|---|
| IMERG | **3.84 mm (48 h)$^{-1}$** | **3.86 mm (48 h)$^{-1}$** | -6.2 % | + 37.0 % |
| RADKLIM | - | - | **6.93 mm (48 h)$^{-1}$** | - |
| IDAWEB | - | - | - | **9.58 mm (48 h)$^{-1}$** |
| COSMO 12 (90 s, PAR) | -37.6 % | -22.0 % | -53.1 % | -29.5 % |
| COSMO 12 (90 s) | -26.3 % | +0.0 % | -0.8 % | -16.0 % |
| COSMO 12 (40 s) | -26.6 % | +2.8 % | +9.4 % | -4.4 % |
| COSMO 4.4 (40 s) | -26.9 % | -2.4 % | -11.6 % | -22.6 % |
| COSMO 4.4 (20 s) | -28.2 % | -2.0 % | -24.3 % | -30.0 % |
| COSMO 2.2 (20 s) | -27.5 % | -1.8 % | -6.9 % | -29.2 % |
| COSMO 2.2 (10 s) | -29.2 % | -3.7 % | -17.8 % | -9.8 % |
| IFS 9 (450 s, PAR) | -17.1 % | +5.6 % | -29.2 % | +17.6 % |
| IFS 9 (450 s) | -4.9 % | +28.0 % | -2.1 % | +73.4 % |
| IFS 9 (240 s) | -0.9 % | +35.4 % | +16.4 % | +86.2 % |
| IFS 4.5 (240 s) | -1.1 % | +34.5 % | +7.4 % | +50.9 % |
| IFS 4.5 (120 s) | -0.6 % | +37.3 % | +11.9 % | +31.2 % |
| IFS 2.9 (120 s) | +0.3 % | +40.1 % | +14.3 % | +37.2 % |
| IFS 2.9 (60 s) | +0.1 % | +40.4 % | +13.4 % | +28.7 % |

[a] Interpolated on COSMO 2.2 km grid

[b] Interpolated on RADKLIM 1 km grid

[c] Nearest-neighbor interpolation to IDAWEB stations

### 3.1.4 Total precipitation

Total precipitation during the two days has been analyzed for four domains: The whole European domain, the land part of the European domain, the RADKLIM domain, and the IDAWEB stations. The results are summarized in Table 2.

For the whole European domain, all COSMO runs show clearly less precipitation than IMERG. And while all COSMO 395 simulations with explicit deep convection produce about the same amount of precipitation, the one with parameterized deep convection is clearly an outlier with even less precipitation. The IFS runs with explicit deep convection show about the same amount of precipitation as IMERG. Also here, the run with paramaterized deep convection shows significantly less precipitation than the explicit ones, similar to the results of the global simulations with IFS by Dueben et al. (2020). This is most probably due to the missing medium-to-heavy precipitation in the parameterized runs as shown in Sect. 3.1.2.

If one looks only at the precipitation over land, COSMO is much closer to IMERG, while the values from IFS are clearly larger. The effect of parameterized deep convection for both models is the same as for the whole European domain but even more distinct as the larger part of deep convection is happening over land. Moreover, IFS shows a clear sensitivity to timestep with the amount of precipitation increasing with decreasing timestep.



One of the properties of hydrostatic systems is supposed to be the overestimation of convective precipitation amount and

area compared to nonhydrostatic systems (Kato and Saito, 1995; Kato, 1997). When looking at total precipitation over land and comparing it with values from IMERG, it looks like IFS is overestimating convective precipitation. Additionally, the overestimation seems to get worse with increasing resolution which is consistent with findings by Kato (1997). However, it is not clear whether this effect can be purely attributed to the hydrostatic core, as there are other factors, notably the subgrid-scale parameterizations, to consider.

Total precipitation in the RADKLIM domain and at the IDAWEB stations has to be interpreted cautiously as both domains are rather small and the simulations cover only 48 hours. But the numbers support the findings from the European domain in the sense that IFS seems to overestimate precipitation while COSMO generally underestimates it. Also, the precipitation-reducing effect of parameterized deep convection is visible for both domains.

### 3.1.5 Cumulative frequency of vertical winds

Figure 6 shows the cumulative frequencies of vertical wind velocity at 850 hPa and 500 hPa. The frequencies were calculated from the instantaneous vertical velocities at every full hour during the 48 hour simulation period. While the distribution on the 850 hPa level is almost symmetric, the updrafts at the 500 hPa become much stronger than the downdrafts. This property is consistent with a cross section of a multicell thunderstorm produced by COSMO in Fig. 7. At 850 hPa, the updrafts are not really that strong due to the proximity to the planetary boundary layer. At 500 hPa, we are well above the level of free

convection and the updraft velocities become very high in this area. In general, downdrafts are more frequent than updrafts in both models and on both levels, but they never develop the strength of the deep convective updrafts. Especially in IFS, the values for the downdrafts are quite low.

The profound impact of deep convection parameterization on the vertical motions in the atmosphere can be seen on the two panels on the left-hand side of Fig. 6. Both the updraft and downdraft velocities are much smaller with parameterized

deep convection. This effect is most pronounced for the IFS updrafts at 500 hPa where the parameterization leads to very low updrafts which is also consistent with the lack of heavy precipitation in this configuration (see Fig. 3).

Both models show some sensitivity to horizontal resolution and the updraft velocities at 500 hPa are also comparable between the respective horizontal resolutions. The timestep sensitivity seems to be more pronounced in IFS at both levels, which we interpret as resulting from the larger vertical motion in combination with a large time step. Nevertheless, the updraft velocities

for the 4.5 km and 2.9 km runs of the hydrostatic IFS are similar to those of the nonhydrostatic COSMO runs with 4.4 km and 2.2 km grid spacing, respectively. Hence, the presumption that the vertical velocities could become unrealistically high due to the violation of the hydrostatic assumption at these resolutions cannot be confirmed. It is not clear to what extent the large timestep of IFS influences these results, but results from Dueben et al. (2020) even show a reduction of updraft velocities with the hydrostatic version of IFS at 1.45 km horizontal grid spacing, when reducing the timestep from 120 s to 30 s.

Another interesting aspect is the disparity in downdraft velocities between IFS and COSMO. The downdraft velocities in IFS are significantly lower for the explicit runs compared to the corresponding COSMO runs. At the same time the probability of having a downdraft is higher in IFS than in COSMO. The lower downdraft velocities in IFS could be related to the hydrostatic





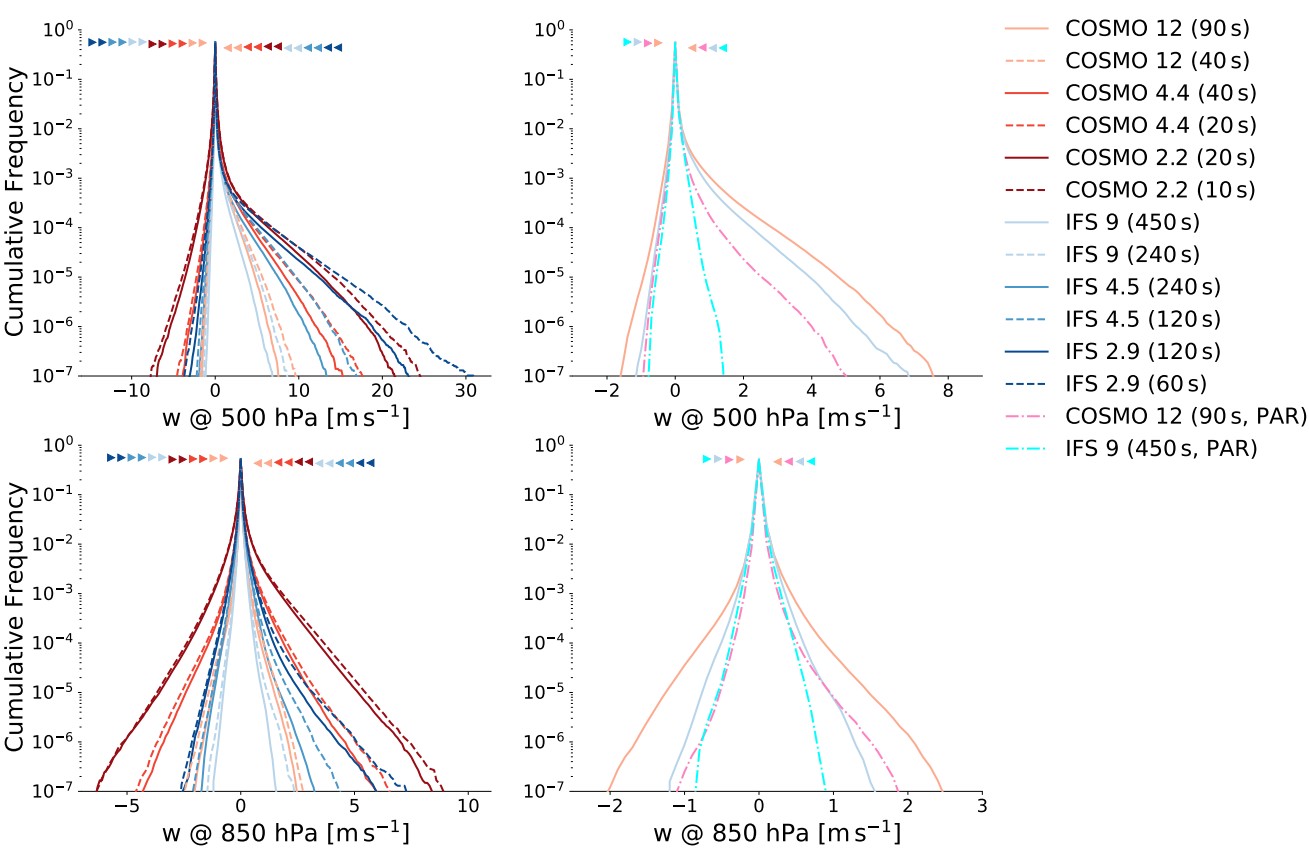

**Figure 6.** Cumulative frequency of vertical wind velocity on the 500 hPa (upper row) and 850 hPa pressure levels (lower row) with the same layout as in Fig. 3. The triangles indicate the height of the respective starting points, that is the frequency of downdrafts ($< 0\,\mathrm{m\,s^{-1}}$) and updrafts ($> 0\,\mathrm{m\,s^{-1}}$). Note the different scales on the horizontal axes.

formulation of the governing equations, as the results from Dueben et al. (2020) generally show higher downdraft values when switching to the nonhydrostatic implementation of IFS. But also diffusion seems to play a critical role in downdraft velocities and might explain the differences between the two models, as shown in Sect. 3.2.

### 3.1.6 Energy spectra

While kinetic energy spectra are generally not used as a measure of a model's skill, they can be useful in order to determine whether a model is able to reproduce the observed dynamics of the atmosphere (Skamarock, 2004). Observational analysis from Nastrom and Gage (1985) showed a transition of the kinetic energy spectra from a $k^{-3}$ dependence at the large scale, characteristic of two-dimensional turbulence, to a $k^{-5/3}$ dependence at the mesoscale, with $k$ being the wavenumber. These results have been confirmed by other studies such as Lindborg (1999); Cho et al. (1999a, b). The two upper panels of Fig. 8 show the Power Spectral Density (PSD) of horizontal kinetic energy $E_{kin}$ for all model runs. With deep convection parameterization





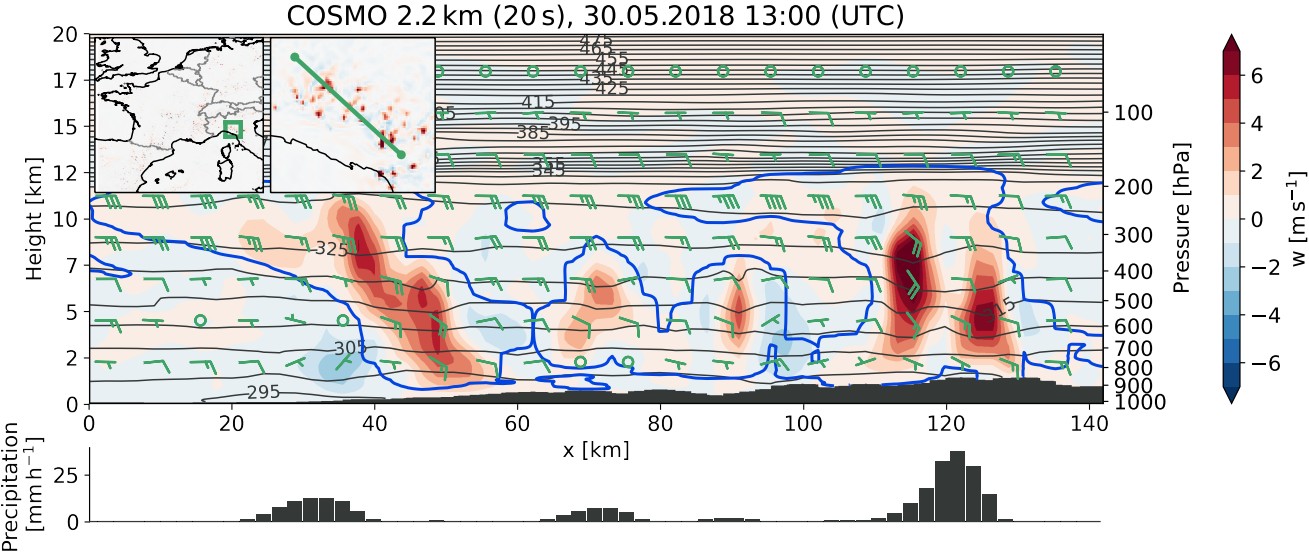

**Figure 7.** Cross section of convective cells over Northern Italy produced by COSMO 2.2 km. The shading represents the vertical wind speed, the green barbs the overall wind direction and velocity, and the blue contour the clouds (cloud water + cloud ice content > 0.01 g kg$^{-1}$). The black contours represent potential temperature and the bar chart at the bottom shows the location and intensity of accumulated precipitation over the last hour.

off, both models are able to nicely reproduce the expected slopes. The amplitude and the location of the transition zone at around 400 km wavelength are almost identical for all explicit simulations and also agree very well with results from Skamarock

and Klemp (2008) who performed numerical studies using the Advanced Research WRF (ARW) model. Deep convection parameterization is clearly influencing the dynamics in the sense that the $k^{-5/3}$ dependence in the mesoscales cannot be reproduced. For horizontal kinetic energy at 500 hPa, both models show very little dependence on timestep, similar to the results from Dueben et al. (2020). COSMO seems to conserve a bit more kinetic energy at smaller wavelengths while IFS shows stronger dissipation at these scales. Malardel and Wedi (2016) examined kinetic energy spectra produced by IFS and

identified subgrid-scale paramterizations (notably surface drag and momentum vertical mixing) as the major contributors to dissipation in IFS. They have also found that differences in orographic filtering will affect the energy transfer. It is likely, that the differences between COSMO and IFS in dissipation rate for smaller wavelengths are caused by a combination of different factors.

So while timestep seems to have little influence on the horizontal kinetic energy spectra, it certainly has an influence on the

vertical wind spectra, as the lower right panel of Fig. 8 shows. For most runs a reduction in timestep leads to significantly more energy throughout all wavelengths. This is most pronounced for the pairs of simulations with larger timesteps, but even for the runs with smaller timesteps it leads to an increase in power, especially for the smaller wavelengths. Similar to the horizontal kinetic energy spectra, COSMO conserves more energy in the smaller wavelengths than IFS. The effect of parameterizing





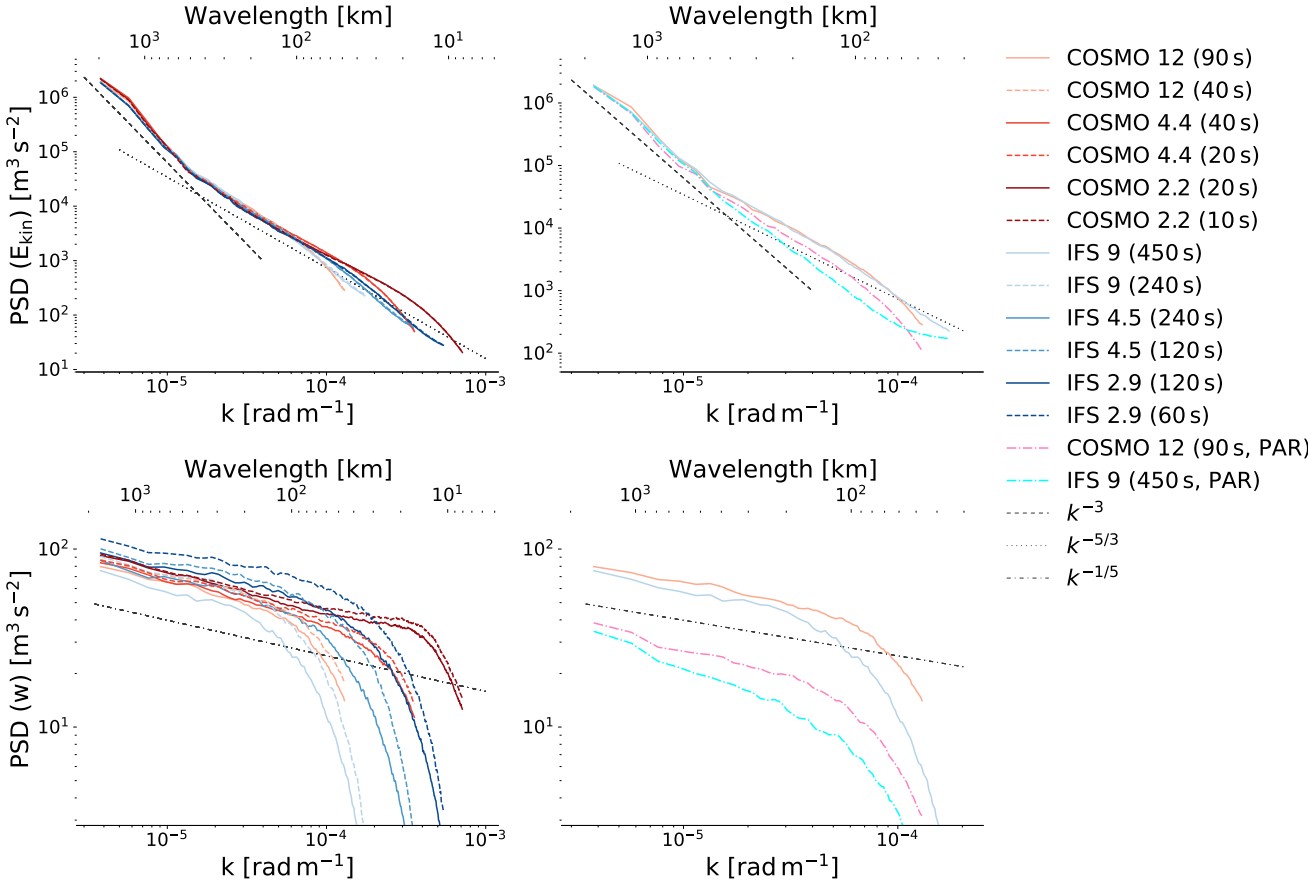

**Figure 8.** Power Spectral Density (PSD) plots for horizontal kinetic energy and $w$ at $500\,\text{hPa}$ (top and bottom panels). The left-hand panels show PSDs of all runs with explicit deep convection. The right-hand panels show PSDs for the coarser runs with parameterized (PAR) and explicit deep convection. Data points for wavelength smaller than $4\Delta x$ or larger than $L/2$, where $L$ is the domain width, have been cut off. Note the different scales on the vertical and horizontal axes.

deep convection seems to be even more drastic for the vertical than for the horizontal winds: The amplitude is clearly reduced
throughout all wavelengths for both models.

The amplitude and shape of the power spectral densities of $w$ from the explicit runs seem to mostly agree with values from other observational and numerical studies (Bacmeister et al., 1996; Gao and Meriwether, 1998; Callies et al., 2016; Schumann, 2019; Panosetti et al., 2019). The spectra of the configurations used in this study all follow a slope of roughly $k^{-1/5}$. This seems to be a realistic value (Liu, 2019), even though there is quite a bit of variability in the aforementioned studies, indicating
that probably there is some dependence on the specific weather situation, altitude, and regional climate considered.



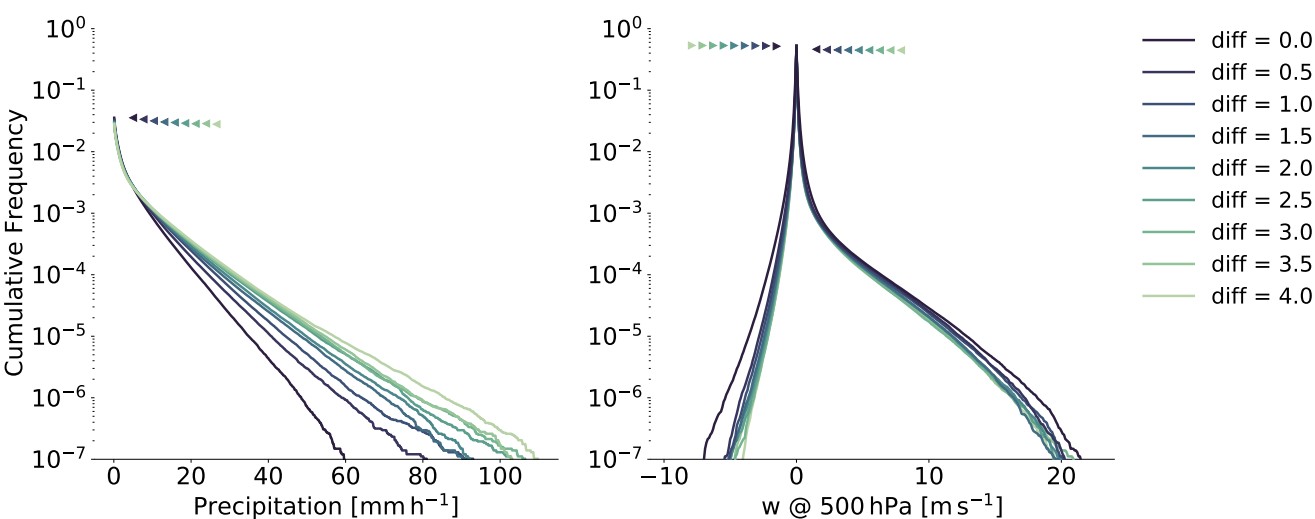

**Figure 9.** Cumulative frequency of hourly precipitation (left) and vertical wind speed at $500\,\mathrm{hPa}$ (right) for runs with COSMO $2.2\,\mathrm{km}$ and different values for explicit horizontal diffusion of wind, temperature, pressure, and moisture variables.

### 3.2 The effect of horizontal diffusion

Figure 9 shows the effect of horizontal diffusion on precipitation and vertical velocity in COSMO. Rather counter-intuitively, additional explicit diffusion leads to a decrease of light precipitation and a significant increase of heavy precipitation. This increase in heavy precipitation events could explain some of the difference between the COSMO $2.2\,\mathrm{km}$ and IFS $2.9\,\mathrm{km}$

simulations shown in Fig. 3, where IFS with $\Delta t = 60\,\mathrm{s}$ also reaches values of $100\,\mathrm{mm\,h^{-1}}$ and more.

The impact of horizontal diffusion on vertical wind velocities is not as prominent as for precipitation, but nevertheless there is a clear pattern. Both updraft and downdraft velocities are reduced with more horizontal diffusion. The relative change is most noticeable for the downdrafts, again with the diffusive configurations showing some similarities to the behavior of IFS in Fig. 6. So increased horizontal diffusion could at least be a possible cause of the, compared to COSMO, low downdraft velocities

in IFS at the $500\,\mathrm{hPa}$ level. The additional diffusion does not really change the downdraft velocities on the $850\,\mathrm{hPa}$ level (not shown) but reduces the updraft velocities and brings them closer to the corresponding profiles from IFS.

Figure 10 shows a snapshot of vertical wind velocities at $500\,\mathrm{hPa}$ and precipitation from a multicell thunderstorm over the area around the Netherlands. Without explicit horizontal diffusion, the cells in COSMO are much smaller than the ones in IFS. But with a large amount of explicit horizontal diffusion, the cells look quite similar to the ones in IFS in terms of size and

shape. The relatively large horizontal extent of convective cells in IFS could be a reason why the hydrostatic approximation still seems to work quite well even at a horizontal grid spacing of $2.9\,\mathrm{km}$. So one could argue that the rather diffusive properties of the model prevent it from entering into a nonhydrostatic regime, where the vertical extent of buoyant cells becomes larger than their horizontal extent.



This observed increase of convective cell size and heavy precipitation with additional horizontal diffusion is very similar the results in Ricard et al. (2013) with AROME. Malardel and Ricard (2015) increased diffusion in the convergent parts of the flow and reduced it in the divergent parts in order to improve the conservation property of the scheme used in IFS, AROME, and HARMONIE, which lead to a decrease of heavy precipitation. Unlike in Malardel and Ricard (2015), the additional diffusion is applied everywhere in our experiment. It is not clear how the results would change with COSMO, if for example diffusion would be applied only to the convergent or divergent part of the flow, but answering that is beyond the scope of this study and would require further investigations.

One of the most important conclusions from the COSMO diffusion experiments is the evidence that horizontal diffusion in the governing equations does not act to simply smooth the precipitation field (which would weaken and broaden all cells, but not significantly change their number). Rather it appears that diffusion more fundamentally affects the dynamics: With higher diffusion, the available CAPE is consumed by substantially fewer but broader updrafts (Fig. 10). In terms of peak vertical velocity, however, the cells weaken, and one wonders why the peak hourly precipitation rates increase so strongly (Fig. 9). We think that the increase of peak precipitation is owed to an accumulation effect. As the cells are much broader, the precipitation footprint at the surface will take longer while propagating over an affected gridpoint. Evidence for such accumulation effects can be seen in the elongated hourly precipitation signatures in Fig. 10.

Figure 11 shows the power spectral densities of the COSMO diffusion experiments, and it is obvious that more diffusion leads to more dampening near the short-wave cut off. In fact, the spectra from COSMO with substantial explicit diffusion look quite similar to the ones obtained from IFS with 2.9 km grid spacing in Fig. 8.

**Figure 10.** Vertical wind at 500 hPa (upper row) and accumulated hourly precipitation (lower row) over the Netherlands on 29 May 2018 at 14:00 UTC. The first column shows the values from COSMO 2.2 km without explicit horizontal diffusion while the middle column shows the results from a simulation with additional diffusion. The right column shows to the values obtained from the IFS 2.9 km simulation with $\Delta t = 120$ s.

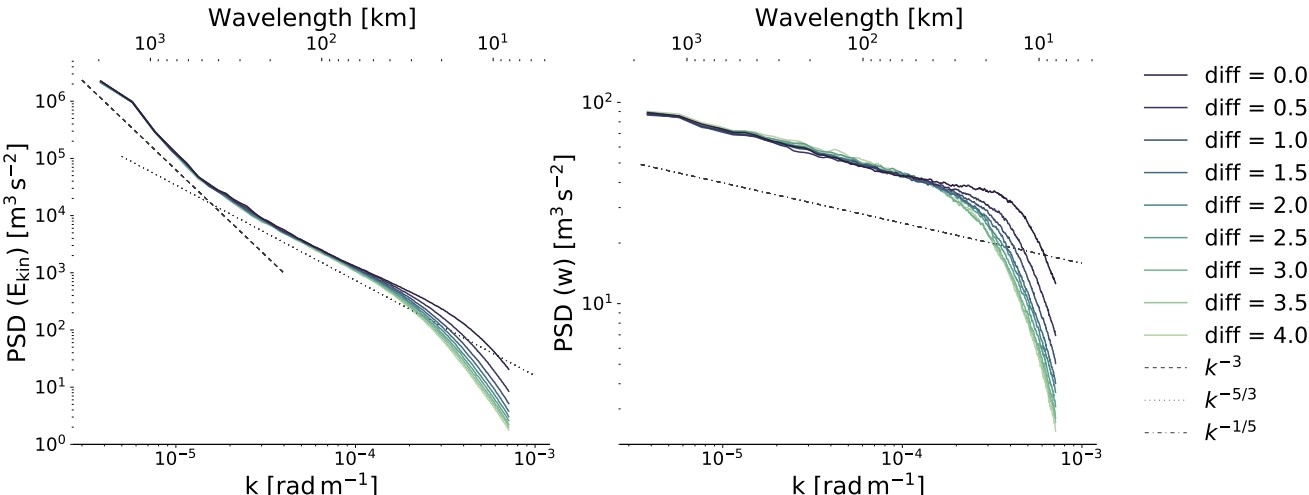

**Figure 11.** Power Spectral Density (PSD) plots for horizontal kinetic energy and $w$ at $500\,\text{hPa}$ for runs with COSMO $2.2\,\text{km}$ and different values for explicit horizontal diffusion of wind, temperature, pressure, and moisture variables. Note the different scales on the vertical axes.

## 4  Conclusions

IFS produces more light precipitation than COSMO in all configurations and generally produces more precipitation. For both models, parameterized deep convection leads to more light precipitation but less medium-to-heavy precipitation. With explicit deep convection, the cumulative frequencies in COSMO are quite constant with regards to horizontal resolution and timestep. This is not the case for IFS, which shows an increasing amount of heavy precipitation with increasing resolution. However, the deciding factor for the precipitation frequencies in IFS seems to be the timestep. IFS runs with a smaller timestep all lead to significantly more heavy precipitation than the respective runs with larger timestep. It is not entirely clear how much this behavior is an effect of timestep on resolved dynamics or the subgrid-scale parameterizations and their coupling. It is possible that a combination of these factors contribute to this timestep sensitivity of precipitation intensities.

The comparison of model results with the three observational datasets IMERG, RADKLIM, and IDAWEB showed that both model's runs with explicit deep convection seem to be in the range of realistic values when it comes to precipitation intensities. In contrast, both runs with parameterized deep convection failed to reproduce the medium-to-heavy precipitation that could be observed during these two days and thus also produced significantly less precipitation.

Resolution and timestep size also have an effect on the diurnal cycle of precipitation over land. A higher spatial and temporal resolutions seems to lead to an earlier onset and peak of precipitation. While we see a convergence of the diurnal cycle already at $4.4\,\text{km}$ grid spacing in COSMO, IFS only shows signs of convergence at the highest resolution with $2.9\,\text{km}$ grid spacing, most probably still due to the relatively large timestep sizes of $120\,\text{s}$ and $60\,\text{s}$. Furthermore, this study also reinforces the evidence that parameterized deep convection leads to a much earlier onset and peak in the diurnal cycle. However, besides the



two coarsest runs (COSMO 12 km and IFS 9 km) with explicit deep convection, all runs seem to have a too early phase in the diurnal cycle when compared with observations from the multi-satellite product IMERG.

The redistribution of heat and moisture due to parameterized deep convection has a distinct effect on the vertical velocities, leading to lower values for the downdrafts and especially the updrafts. From the runs with explicit deep convection, the respective updraft values at the 500 hPa level were quite similar between the nonhydrostatic COSMO and the hydrostatic IFS.

This indicates that the hydrostatic approximation at a grid spacing of around 2-3 km still works well and does not lead to too high updraft values. However, the downdraft values in IFS are significantly lower than in COSMO throughout almost all simulations. This could be a characteristic of hydrostatic models (see for example Dueben et al., 2020) or also be caused by enhanced diffusion in IFS compared to COSMO. However, this would require further investigation.

The influence of timestep on wind velocities does not seem to be very crucial for the horizontal winds and both models show

almost no change in the spectra of horizontal kinetic energy with different timesteps. The vertical winds however, are clearly influenced by the timestep. This is visible in changes in spectra and also frequency distributions where a large timestep seems to suppress high vertical velocities. The importance of resolving all these high velocities, compared to the significant additional computational costs involved with a smaller timestep, is up for debate and probably also depends on application and purpose of the simulation.

Increasing horizontal diffusion in COSMO leads to more medium and heavy precipitation, making the precipitation frequency profile in this range look similar to the ones from IFS. Furthermore, more horizontal diffusion also leads to a reduction of downdraft velocities at the 500 hPa level and thus also making the vertical velocity profiles of COSMO and IFS more akin. The added diffusion generally leads to fewer convective cells while increasing the horizontal extent of these cells. This could be a reason why the hydrostatic approximation still seems to work quite well even at a grid spacing of around 2-3 km, as the

relatively large horizontal width of the cells might prevent them from entering the nonhydrostatic regime where the vertical extent of the buoyant cells becomes larger than the horizontal extent. But while this sensitivity to dissipation certainly would need a more detailed investigation, it seems to explain some of the characteristic differences between COSMO and IFS.

*Code and data availability.*   Model codes developed at ECMWF are the intellectual property of ECMWF and its member states, and therefore the IFS code is not publicly available. Access to a reduced version of the IFS code may be obtained from ECMWF under an OpenIFS licence

(see http://www.ecmwf.int/en/research/projects/openifs for further information, last access: 28 January 2021). The particular version of the COSMO model used in this study is based on the official version 5.0 with many additions to enable GPU capability and available under license (see http://www.cosmo-model.org/content/consortium/licencing.htm for more information, last access: 28 January 2021). Most of these developments have been reintegrated into the mainline COSMO version in the meantime. COSMO may be used for operational and for research applications by the members of the COSMO consortium. Moreover, within a liense agreement, the COSMO model may be used

for operational and research applications by other national (hydro-)meteorological services, universities, and research institutes. ECMWF operational analysis data, which has been used for initial (IFS and COSMO) and lateral boundary conditions (COSMO), is available at https://www.ecmwf.int/en/forecasts/dataset/operational-archive (last access: 28 January 2021). The model output data from IFS and COSMO used for the figures in this work, as well as the intial conditions for the soil in COSMO are available under https://doi.org/10.5281/zenodo.4479130.



The rain gauge data over Switzerland can be accessed from the IDAWEB web portal at MeteoSwiss (https://gate.meteoswiss.ch/idaweb, last

access: 28 January 2021). The RADKLIM dataset is available under https://opendata.dwd.de/climate_environment/CDC/grids_germany/ hourly/radolan/reproc/2017_002/bin/ (last access: 28 January 2021) and the GPM IMERG dataset is available under https://gpm1.gesdisc. eosdis.nasa.gov/data/GPM_L3/GPM_3IMERGHH.06 (last access: 28 January 2021).

*Author contributions.* CZ, NPW, and CS designed the experiments. CZ performed the COSMO model simulations and NPW the corresponding IFS simulations. CZ performed the analysis of model output and observations with supervision from CS and NB, and technical support

from PDD and NPW. NPW, PDD, and CS were strongly involved in the discussion of the results. CZ wrote the paper with input from all other co-authors.

*Competing interests.* The authors declare that they have no conflict of interest.

*Acknowledgements.* We acknowledge PRACE for awarding compute resources for the COSMO simulations on Piz Daint at the Swiss National Supercomputing Centre (CSCS). We also acknowledge the Federal Office for Meteorology and Climatology MeteoSwiss, CSCS,

and ETH Zurich for their contributions to the development of the GPU-accelerated version of COSMO. We would like to thank Elmar Weigl and Marcus Paulat (DWD) for their assistance concerning the RADKLIM dataset, and Pirmin Kaufmann (MeteoSwiss) for his help regarding the IDAWEB observations. Peter D. Dueben gratefully acknowledges funding from the Royal Society for his University Research Fellowship and the ESIWACE2 project. ESIWACE2 has received funding from the European Union's Horizon 2020 research and innovation programme under grant agreement No. 823988.



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
