# Peer review of "Model intercomparison of COSMO 5.0 and IFS 45r1 at kilometer-scale grid spacing"

_Geoscientific Model Development, 2021_

## Referee Comment (RC2)

Review of: Model intercomparison od COSMO 5.0 and IFS 45r1 at kilometre scale grid spacing

By: Zeman et al.
Overall Judgment:  Minor Revisions

**General Comments**

This paper is a good example of a sensitivity study of 2 different models that operate at the kilometre scale. I am less sure the paper is very useful as a intercomparison as the differences between the models are so huge that it has turned out difficult to pinpoint physical and numerical reasons for the differences in simulation results between the 2 models. Of course many of the results beg for additional numerical experiments and I completely understand that these are beyond the scope of this paper. Having said that there is one sensitivity run that is strangely missing and that is a non-hydrostatic run of the IFS: One of the intriguing questions of this paper is at which resolution it is appropriate or even mandatory to switch to a non-hydrostatic formulation. In that spirit at least one non-hydrostatic run of the IFS would have so logical to include, even if the differences with the hydrostatic version would remain small, even for the highest resolution. So inclusion of results of at least the non-hydrostatic 2.9km resolution IFS would be strongly recommended.

**Further Comments**

- **Concerning the observational data sets:** I was under the impression that through OPERA there is already a European rainradar network that provides a (West) - European coverage, instead of being dependent on the radar data of only one European country. Perhaps the authors can comments on this

- **Shallow Convection on/off**:. I would be interested if the authors could say something on the results where also the shallow convection is switched off . I ask this since I believe that COSMO at 2 km in general runs without a shallow convection parameterisation, In that sense it would be good to know the performance of COSMO in its "operational' setting.

- **Deep Convection on/off:** If the authors believe that the skill of both models with respect precipitation is in general better without the deep convection parameterisation,, even at coarser resolution, as also claimed by several other studies, could they please comment on the reasons why most operational models keep running with a (deep) convection parameterisation at these resolutions.

- **Downdrafts**: Why are the simulated up- and downdrafts at 850 mb symmetric as we know they are not in reality. Is this related that the resolution is not fine enough to resolve this asymmetry? Why are the too weak downdrafts of the IFS be related to the hydrostatic formulation? ( line 438) What could be a plausible physical reason?

- **Horizontal Diffusion:**  The shift to stronger precipitation  rates with larger horizontal diffusion in COSMO  is indeed counter-intuitive, especially when realising that the updraft strengths are decreasing at the same time. What could be an explanation for this. Is there a similar sensitivity in the IFS? With which strength of horizontal diffusion of COSMO can the strength of horizontal diffusion used in the IFS be compared? How do these values compare with a more physical Smagorinski based horizontal diffusion strength? I presume

that would suggest a lower strength for horizontal diffusion of the IFS and prompts the question why the IFS is using a relatively strong horizontal diffusion strength. It would important to know the answer to this question since it is suggested in this paper that the reason that IFS can still run hydrostatically is possibly related to the fact that it is using such a strong horizontal diffusion which reduces the difference between hydrostatic and non-hydrostatic formulation.

- **Fig 10:** This figure does not really seem to support the results in figure 9 as I essentially se less areas with precip in excess of 1.5 mm/hr.

- It would be nice to have a list of suggested follow-up sensitivity simulations for further studies such as : separate varying the time stepping for the physics and the dynamics. Sensitivity for horizontal diffusion in the IFS, effect of shallow convection parameterisation etc…

**Typos**

line 319 : are => is

---

## Author Comment (AC1)

**Answers to Reviewer Comments 1:**

Many thanks for reviewing our paper and the clear and constructive comments on how we could improve it. We have tried to incorporate most suggestions. See below for more detailed answers on the specific comments.

Review of "model intercomparison of COSMO and IFS by Zeman et al

This paper studies the behavior of two popular NWP models on the kilometer type scale, and explores the effects of hydrostaticity, time step, and spatial resolution. A somewhat confounding issue is that the two models are quite different in set up, and the results remain therefore somewhat anecdotal. However, I still think that the paper would be an interesting addition to the literature. I only have a few simple points that I would ask the authors to look at:

Fig1: Eyeballing, one could argue that IFS 9/450 did the best, at least in terms of cluster size and location. A lot is also dominated by the differences between IFS and COSMO. Is there any simple metric that could quantify this?

Even though it would be interesting to include some sort of performance score, we would like to avoid it, because the paper is not meant to be a performance comparison of the two models. We also think that the scores for such a relatively small sample size would not be very meaningful and could distract the reader from the main findings. In addition, the version of the COSMO model employed is normally used as a regional climate model, and not as a NWP forecasting model.

We have expanded the section regarding cluster size and location with the following text at line 317 in the updated manuscript:

"Looking at Fig. 1, one could argue that IFS 9 km (450 s) without deep convection parameterization is the closest to IMERG with regards to cluster size and location. However, this is a momentarily snapshot and while the IFS 9 km operational configuration is well-tuned, there is evidence of shortcomings and ongoing work to improve cluster size and location in the IFS model with deep convection parameterization."

L95: While numerically stable, there is no guarantee that the model would produce reasonable results at a CFL of 4. Some foreshadowing would be good here.

We agree, and while the cited paper by Staniforth and Côté (1991) shows some evidence, this is mostly for idealized cases. Therefore we have added the following sentence to the new manuscript (line 96):

"Evidence that the semi-Lagrangian semi-implicit scheme does not only provide numerical stability for such high CFL numbers, but also produces reasonable results in real-world scenarios is provided by the day-to-day forecasts of ECWMF, using a competitive timestep of 450 s at 9 km grid with IFS."

L320: If that were the reason, a timestep effect would quickly saturate out and not show up for the higher res IFS simulations, right? Would a smaller timestep in the subgrid model help as well?

ECMWF does indeed see significant sensitivity using different timesteps for physics and dynamics in the development of their alternative dynamical core (FVM). However, we think that a detailed study of this is beyond the scope of this paper. We would also have to consider that the vertical velocities will probably increase (up to some point) with higher horizontal resolutions.

We have expanded the section by mentioning possible convergence studies with changing the timestep of the dynamics and physics independently with the following text (see line 367 in the updated manuscript):

"For example, to quantify the effect of (a), one could perform a convergence analysis with IFS by changing the timestep for the dynamics while keeping the timestep for the subgrid-scale parameterizations constant. Similarly, to quantify the effect of (b), the timestep for the subgrid-scale parameterizations could be changed while keeping the timestep for the dynamics constant. However, this is beyond the scope of this work."

Fig 10 shows some gravitational waves initiated by the convection. The differences between the diff=0 and 4 versions are stark; are these waves (not just the convection cells!) physical?

MODIS satellite pictures from a similar time at 12:50 UTC in this region showed similar circular patterns in cloudy/cloud-free areas around the large convective cells (see image below). Therefore, we would consider the waves as physical and likely associated with propagating cold air pools. However, we feel that investigating and validating this behavior in more detail would require a significant expansion of an already rather long paper and therefore we consider it beyond the scope of this paper.

[Figure]

AQUA MODIS
12:50 UTC
2019-05-29
(Source: https://worldview.earthdata.nasa.gov/)

Smaller points:

L33: Are these essentially poorly resolved gridpoint storms? That's relevant here, because a 1.5km simulation would still not resolve the deep convection, so still result in grid point storms/grey zone issues.

According to the cited literature on line 34/35, the overestimation of precipitation frequency and underestimation of intensity is mainly an effect of the parameterization leading to a premature initiation of convection and thus preventing a realistic accumulation of CAPE which would lead to heavier and more concentrated precipitation. And yes, the grid point storms/grey zone issues still exist at kilometer-scale resolution where we are not able to fully resolve deep convection.

L219: "by" default; "the" deep convection parameterization on

Changed. Thank you!

L227: How does this work as a source term? I'm having difficulties with the units. Writing it out explicitly might be helpful.

We have added the equation and a sentence describing the different terms (see lines 237 and following in the new manuscript).

Fig5: The spaghetti plots are hard to decipher, and most of what I want to get out of it is a peak intensity and its time for each simulation. Perhaps a scatter plot (with abbreviations instead of dots) may make more sense?

We agree that it's not the easiest plot to read. However, next to the peak, onset and length of diurnal precipitation is also of interest to us and we see clear differences there between the two models and also the observations. Therefore, we think that this line plot gives a more complete picture.

**Answers to Reviewer Comments 2:**

We thank the reviewer for reviewing the paper and the good comments and suggestions. We have tried to address the comments and changed some parts of the manuscript according to them. Please see below for more details.

**General Comments**

This paper is a good example of a sensitivity study of 2 different models that operate at the kilometre scale. I am less sure the paper is very useful as a intercomparison as the differences between the models are so huge that it has turned out difficult to pinpoint physical and numerical reasons for the differences in simulation results between the 2 models. Of course many of the results beg for additional numerical experiments and I completely understand that these are beyond the scope of this paper. Having said that there is one sensitivity run that is strangely missing and that is a non-hydrostatic run of the IFS:  One of the intriguing questions of this paper is at which resolution it is appropriate or even mandatory to switch to a non-hydrostatic formulation. In that spirit at least one non-hydrostatic run of the IFS would have so logical to include, even if the differences with the hydrostatic version would remain small, even for the highest resolution. So inclusion of results of at least the non-hydrostatic 2.9km resolution IFS would be strongly recommended.

We agree with the reviewer that, due to the huge differences between the models, the study is not able to pinpoint the causes for the observed differences in model results. Nevertheless, we believe that the paper can give some indications and represents a reasonable base for further, more detailed studies.

While it would be certainly interesting to also include a run with the nonhydrostatic version of IFS, we feel that this would not significantly strengthen the argument for or against the use of the hydrostatic approximation at 2.9 km grid spacing. Besides the use of the hydrostatic approximation, there are several other major differences between the nonhydrostatic and the hydrostatic version of IFS (most notably the use of a different vertical integration and the iterative timestepping scheme) which would probably open up even more questions and extend the paper. We have added the following paragraph to the model description section of IFS where we directly address this issue (line 201 in the new manuscript) including also the omission of the newly developed nonhydrostatic finite volume version of IFS (IFS-FVM).

"While it would certainly be interesting to include all three different IFS model versions into this intercomparison, the differences between them are substantial and we consider an intercomparison among them beyond the scope of this work. A detailed comparison of the spectral hydrostatic IFS with the spectral nonhydrostatic IFS can be found in Dueben et al. 2020, who performed global simulations at 1.45 km grid spacing with these model versions in different configurations. In the present study, we have decided to focus only on the operational spectral hydrostatic model version of IFS."

We would further like to mention that ECMWF is in fact in the process of making runs with a nonhydrostatic and hydrostatic version of IFS with as close as possible numerics. However, this poses to be quite a challenge (related to an unexpected contribution to differences, namely the physics-dynamics coupling) and is beyond the scope of this paper.

**Further Comments**

**Concerning the observational data sets:**
I was under the impression that through OPERA there is already a European rainradar network that provides a (West) - European coverage, instead of being dependent on the radar data of only one European country. Perhaps the authors can comments on this

We thank the reviewer for the good suggestion. We are in the process of getting an account for accessing the OPERA product and we will most likely use it for future work. For this work, we think that the RADKLIM dataset, which still covers a relatively large area, is still a good candidate as a radar-based observational product.

**Shallow Convection on/off:**
I would be interested if the authors could say something on the results where also the shallow convection is switched off. I ask this since I believe that COSMO at 2 km in general runs without a shallow convection parameterisation. In that sense it would be good to know the performance of COSMO in its "operational" setting.

To our knowledge, most operational COSMO runs around 2 km grid spacing are performed with shallow convection parameterization. The German Weather Service (DWD) uses shallow convection parameterization for their runs with 2.8 km grid spacing (https://www.dwd.de/EN/research/weatherforecasting/num_modelling/01_num_weather_prediction_modells/regional_model_cosmo_de.html). And while MeteoSwiss turn off shallow convection parameterization for their 1.1 km runs, they have shallow convection on for their 2.2 km ensemble runs (personal communication). Vergara-Temprado 2020* performed some 2.2 km COSMO runs without shallow convection but found little difference between the runs with and without shallow convection parameterization. Given this, we would like to avoid extending the paper even more by adding a run (and the accompanying discussion) without shallow convection to the paper.

*https://journals.ametsoc.org/view/journals/clim/33/5/jcli-d-19-0286.1.xml

**Deep Convection on/off:**
If the authors believe that the skill of both models with respect precipitation is in general be\er without the deep convection parameterisation, even at coarser resolution, as also claimed by several other studies, could they please comment on the reasons why most operational models keep running with a (deep) convection parameterisation at these resolutions.

ECMWF continuously and successfully improved its convection parameterization (including deep convection parameterization) over the past 20 years and given its importance for forecast skill for a range of global verification measures it is not that simple to switch this off. This work is part of a wider effort to explore this option or further improvement of the parametrization schemes. Moreover, the 'seamless' model is used in a range of forecast applications including at ECMWF the medium-range (at 9km average grid-spacing), (ensemble at 18km average grid-spacing) and extended-range, seasonal and Copernicus re-analysis and atmospheric composition applications (at 30-40km grid-spacing). There are practical challenges with the seamless modelling strategy that require thought.

**Downdrafts:**
Why are the simulated up- and downdrafts at 850 mb symmetric as we know they are not in reality. Is this related that the resolution is not fine enough to resolve this asymmetry? Why are the too weak downdrafts of the IFS be related to the hydrostatic formulation? ( line 438) What could be a plausible physical reason?

The symmetry could be related to insufficient resolution, especially considering the fact that it becomes less pronounced for the higher resolutions. But there could also be a model specific bias due to the interaction/growth with shallow convection being parameterized, and/or interaction with the equally parametrized boundary layer turbulence. We are also not sure about how far off these profiles are off from reality. Ansmann et al. (2010)* performed doppler lidar measurements in summer 2006 at Leipzig and they observed some asymmetry of the vertical velocities at 1095 m height (Fig. 4), but the observed vertical velocity profile is not highly asymmetric and even a bit less asymmetric than for example the ones obtained from the high-resolution IFS simulations at 850 hPa.

*https://acp.copernicus.org/articles/10/7845/2010/

**Horizontal Diffusion:**
The shift to stronger precipitation rates with larger horizontal diffusion in COSMO  is indeed counter-intuitive, especially when realising that the updraft strengths are decreasing at the same time. What could be an explanation for this. Is there a similar sensitivity in the IFS? With which strength of horizontal diffusion of COSMO can the strength of horizontal diffusion used in the IFS be compared? How do these values compare with a more physical Smagorinski based horizontal diffusion strength? I presume that would suggest a lower strength for horizontal diffusion of the IFS and prompts the question why the IFS is using a relatively strong horizontal diffusion strength. It would important to know the answer to this question since it is suggested in this paper that the reason that IFS can still run hydrostatically is possibly related to the fact that

it is using such a strong horizontal diffusion which reduces the difference between hydrostatic and non-hydrostatic formulation.

As discussed in the manuscript (line 521), we think that the higher frequency of heavy precipitation with additional diffusion is owed to an accumulation effect. As the cells are broader, the precipitation footprint at the surface will take longer while propagating over an affected gridpoint. However, this would have to be verified with additional studies by also looking at instantaneous precipitation rates (which we don't have with the current set of simulations).

A comparison of effective diffusion strength is difficult to achieve without specific idealized experiments due to the different amount of implicit diffusion and the dependency on the horizontal tension and shear of the Smagorinsky diffusion. We believe that the spectras presented in Fig. 8 and Fig. 11 give some idea about the respective dissipative properties of the different configurations of COSMO and IFS. However, we would like to stay away from a general recommendation concerning the optimal diffusion strength, as we think that this would require a much more detailed study which is beyond the scope of this paper.

**Fig 10:**
This figure does not really seem to support the results in figure 9 as I essentially see less areas with precip in excess of 1.5 mm/hr.

The figure does indeed show a bit less precipitation for the COSMO run with additional diffusion, but this is mostly precipitation up to 5 mm/h. The more diffusive COSMO run starts to have higher frequencies at around > 10 mm/h. Below is the same plot with only > 10 mm/h precipitation. This still agrees with the results in Figure 9 since the area is around the same. Another snapshot of another area with more focus on precipitation > 20 mm/h (where the differences really start to show) for the diffusive COSMO run can certainly be found, but we feel that this is a nice plot in terms of showing the generally larger convective cells in the diffusive setup.

[Figure]

It would be nice to have a list of suggested follow-up sensitivity simulations for further studies such as : separate varying the time stepping for the physics and the dynamics. sensitivity for horizontal diffusion in the IFS, effect of shallow convection parameterisation etc.

We agree and have added the following paragraph at the end of the manuscript:

"As already mentioned in the introduction, the comparison of two so fundamentally different models makes it very difficult to confidently attribute disparities in the shown results to specific model properties. While this study is able to give some indications, it also leaves many questions unanswered, which would require more detailed follow-up studies. It would for example be intriguing to only vary the timestep of the dynamics while leaving the timestep for the physical parameterizations constant (or the other way around) in order to get a better idea of the timestep sensitivity of the dynamics and the physics. Another interesting experiment would be to change the amount of explicit diffusion in IFS in order to see whether the effects on precipitation, vertical wind speed, and convective cell size are the same as for COSMO. Moreover, a similar intercomparison between a nonhydrostatic and a hydrostatic model with nearly identical numerics and the same physical parameterizations would give a much better idea about the validity of the hydrostatic approximation at kilometer-scale grid spacing by eliminating as many as possible other causes for differences in model results. This could for example be achieved by trying to bring the nonhydrostatic spectral IFS closer to the operational hydrostatic spectral IFS with regards to the timestepping scheme and the vertical discretization."

Typos
line 319 : are => is

Corrected, thank you!

---

## Author Response (AR2)

**Answer to Topical Editor Comment:**

Comments to the Author:
Thank you for working through the comments. You have addressed well the comments.

As a last task, I would like to ask you to revise the closing paragraph in the conclusions. You give an overview of things that could be done in the future, but in my view you do not use your own results sufficiently.

I would appreciate if you could work in your results a bit more and leave a few hypotheses for testing for future studies rather than only mentioning the topics.

Many thanks for the good comment. We have revised the closing paragraph and included a few results and hypotheses (see the updated manuscript and the latexdiff pdf for details).